Microbiology
Spectrum
# Evaluation of Four Strategies for SARS-CoV-2 Detection: Characteristics and Prospects

Yuqing Chen,[a,b,c] Yu Ma,[a,b,c] Yanxi Han,[a,c] Zhenli Diao,[a,b,c] Lu Chang,[a,b,c] Jinming Li,[a,b,c] Rui Zhang[a,b,c]

aNational Center for Clinical Laboratories, Institute of Geriatric Medicine, Chinese Academy of Medical Sciences, Beijing Hospital/National Center of Gerontology, Beijing, People's Republic of China
bGraduate School of Peking Union Medical College, Chinese Academy of Medical Sciences, Beijing, People's Republic of China
cBeijing Engineering Research Center of Laboratory Medicine, Beijing Hospital, Beijing, People's Republic of China

Yuqing Chen and Yu Ma contributed equally to this article. Author order was determined alphabetically, based on surnames.

**ABSTRACT** The pandemic of severe acute respiratory syndrome coronavirus 2 (SARS-CoV-2) has posed an enormous burden on the global public health system and has had disastrous socioeconomic consequences. Currently, single sampling tests, 20-in-1 pooling tests, nucleic acid point-of-care tests (POCTs), and rapid antigen tests are implemented in different scenarios to detect SARS-CoV-2, but a comprehensive evaluation of them is scarce and remains to be explored. In this study, 3 SARS-CoV-2 inactivated cell culture supernatants were used to evaluate the analytical performance of these strategies. Additionally, 5 recombinant SARS-CoV-2 nucleocapsid (N) proteins were also used for rapid antigen tests. For the wild-type (WT), Delta, and Omicron strains, the lowest inactivated virus concentrations to achieve 100% detection rates of single sampling tests ranged between $1.28 \times 10^2$ to $1.02 \times 10^3$, $1.28 \times 10^2$ to $4.10 \times 10^3$, and $1.28 \times 10^2$ to $2.05 \times 10^3$ copies/mL. The 20-in-1 pooling tests ranged between $1.30 \times 10^2$ to $1.04 \times 10^3$, $5.19 \times 10^2$ to $2.07 \times 10^3$, and $2.59 \times 10^2$ to $1.04 \times 10^3$ copies/mL. The nucleic acid POCTs were all $1.42 \times 10^3$ copies/mL. The rapid antigen tests ranged between $2.84 \times 10^5$ to $7.14 \times 10^6$, $8.68 \times 10^4$ to $7.14 \times 10^6$, and $1.12 \times 10^5$ to $3.57 \times 10^6$ copies/mL. For the WT, Delta AY.2, Delta AY.1/AY.3, Omicron BA.1, and Omicron BA.2 recombinant N proteins, the lowest concentrations to achieve 100% detection rates of rapid antigen tests ranged between 3.47 to 142.86, 1.74 to 142.86, 3.47 to 142.86, 3.47 to 142.86, and 5.68-142.86 ng/mL, respectively. This study provided helpful insights into the scientific deployment of tests and recommended the full-scale consideration of the testing purpose, resource availability, cost performance, result rapidity, and accuracy to facilitate a profound pathway toward the long-term surveillance of coronavirus disease 2019 (COVID-19).

**IMPORTANCE** In the study, we reported an evaluation of 4 detection strategies implemented in different scenarios for SARS-CoV-2 detection: single sampling tests, 20-in-1 pooling tests, nucleic acid point-of-care tests, and rapid antigen tests. 3 SARS-CoV-2-inactivated SARS-CoV-2 cell culture supernatants and 5 recombinant SARS-CoV-2 nucleocapsid proteins were used for evaluation. In this analysis, we found that for the WT, Delta, and Omicron supernatants, the lowest concentrations to achieve 100% detection rates of single sampling tests ranged between $1.28 \times 10^2$ to $1.02 \times 10^3$, $1.28 \times 10^2$ to $4.10 \times 10^3$, and $1.28 \times 10^2$ to $2.05 \times 10^3$ copies/mL. The 20-in-1 pooling tests ranged between $1.30 \times 10^2$ to $1.04 \times 10^3$, $5.19 \times 10^2$ to $2.07 \times 10^3$, and $2.59 \times 10^2$ to $1.04 \times 10^3$ copies/mL. The nucleic acid POCTs were all $1.42 \times 10^3$ copies/mL. The rapid antigen tests ranged between $2.84 \times 10^5$ to $7.14 \times 10^6$, $8.68 \times 10^4$ to $7.14 \times 10^6$, and $1.12 \times 10^5$ to $3.57 \times 10^6$ copies/mL. For the WT, Delta AY.2, Delta AY.1/AY.3, Omicron BA.1, and Omicron BA.2 recombinant N proteins, the lowest concentrations to achieve 100% detection rates of rapid antigen tests ranged between 3.47 to 142.86, 1.74 to 142.86, 3.47 to 142.86, 3.47 to 142.86, and 5.68 to 142.86 ng/mL, respectively.

Address correspondence to Jinming Li, jmli@nccl.org.cn, or Rui Zhang, ruizhang@nccl.org.cn.

The authors declare no conflict of interest.

**KEYWORDS** 20-in-1 pooling tests, SARS-CoV-2 detection, analytical sensitivity, nucleic acid POCTs, rapid antigen tests, scientific deployment, single sampling tests

Since December of 2019, the pandemic of coronavirus disease 2019 (COVID-19) caused by the severe acute respiratory syndrome coronavirus 2 (SARS-CoV-2) has posed an enormous burden on the global public health system has had disastrous socioeconomic consequences (1). As of August 26, 2022, the cumulative number of COVID-19 cases reported worldwide has exceeded 596 million, and the number of deaths has exceeded 6 million (2). A total of 242,307 confirmed cases and 5,226 deaths have been reported in mainland China (3).

Naturally occurring mutations can be selected continuously during virus transmission, which is a dominating obstacle to the prevention and control of COVID-19. Among the variants of concern (VOC) and variants of interest (VOI) designated by the World Health Organization (WHO), based on the transmissibility, pathogenicity, and threat to public health (4), the Delta and Omicron variants were first reported in Guangdong on May 18, 2021 (5), and in Tianjin on December 13, 2021 (6), respectively. The P618R mutation of the Delta variant was found to be closely associated with enhanced viral fusogenicity and pathogenicity (7, 8). The Omicron variant harbors a large number of mutations that have been proven to be involved in higher binding affinity with angiotensin-converting enzyme 2 (ACE2) (9), enhanced pathogenicity and transmissibility (10), and an increased ability of immune evasion (11). In order to confirm infected cases, quarantine populations at risk, and interrupt the chain of transmission, scientific and appropriate detection methods and strategies applicable to specific scenarios should be employed (12).

Real-time reverse transcriptase polymerase chain reaction S(rRT-PCR) is based on the genomic sequences of SARS-CoV-2 and is regarded as the most sensitive and specific method for confirming clinical diagnoses (13). Driven by the throughput limitations of single sampling tests and the urgent demand for improving both the daily testing capacity and the overall testing efficiency in low-risk regions, 5-in-1, 10-in-1, and 20-in-1 sample pooling strategies are proposed and executed for large-scale population screening (14). However, due to the fact that rRT-PCR requires sophisticated equipment, professional personnel, and has a long turnaround time (15), the nucleic acid point-of-care test (POCT) was developed to confer the advantages of high sensitivity, short turnaround time, and independence of laboratory settings (16). Considering the necessity of conducting targeted screening in locations with high risks of transmission, the Chinese government advocates employing rapid antigen tests that usually detect viral nucleoproteins as complementary health tools. As of August 29, 2022, 32 rapid antigen tests have been approved by the National Medical Products Administration (NMPA) for emergency use and are being used in various settings to support rRT-PCR (17).

Regarding the detection methods mentioned above, analytical sensitivity is a fundamental parameter that can facilitate decisions for application and determine their practical use in various scenarios. The performance of the NMPA-approved rRT-PCR assays has been comprehensively analyzed and demonstrated previously; however, there are restricted data on the performance characteristics of these newly approved rapid antigen tests. Additionally, the evaluation of single sampling tests, 20-in-1 pooling tests, nucleic acid POCTs, and rapid antigen tests implemented in different scenarios is scarce and remains to be explored. Therefore, this study aimed to evaluate the analytical sensitivities of these four detection strategies applied in different settings and to provide sufficient research-based evidence for the scientific refinement of the deployment of these tests.

## RESULTS

**Nucleic acid amplification tests.** In this study, 50 $\mu$L of inactivated cell culture supernatants were added to different volumes of sample preservation solution or extraction solution, and the final concentration obtained was used for analysis. For single sampling tests and 5 rRT-PCR detection kits, the lowest inactivated virus concentrations to achieve 100%

**TABLE 1** Calculation of the predilution ratio of different detection methods

| Detection kits | Volume of original samples | Sample preservation solution/sample extraction solution[a] | Final volume | Predilution ratio |
|---|---|---|---|---|
| Real-time RT-PCR tests | | | | |
| Kit N01 | 50 $\mu$L | Single sampling test, 3 mL; 20-in-1 pooling test, 12 mL | Single sampling test, 3.05 mL; 20-in-1 pooling test, 12.05 mL | Single sampling test; 61; 20-in-1 pooling test, 241 |
| Kit N02 | 50 $\mu$L | | | |
| Kit N03 | 50 $\mu$L | | | |
| Kit N04 | 50 $\mu$L | | | |
| Kit N05 | 50 $\mu$L | | | |
| | | | | |
| Point-of-care test | | | | |
| Kit P01 | 50 $\mu$L | 500 $\mu$L | 550 $\mu$L | 11 |
| | | | | |
| Rapid antigen tests | | | | |
| Kit A01 | 50 $\mu$L | 400 $\mu$L | 450 $\mu$L | 9 |
| Kit A02 | 50 $\mu$L | 500 $\mu$L | 550 $\mu$L | 11 |
| Kit A03 | 50 $\mu$L | 500 $\mu$L | 550 $\mu$L | 11 |
| Kit A04 | 50 $\mu$L | 340 $\mu$L | 390 $\mu$L | 7.8 |
| Kit A05 | 50 $\mu$L | 300 $\mu$L | 350 $\mu$L | 7 |
| Kit A06 | 50 $\mu$L | 300 $\mu$L | 350 $\mu$L | 7 |
| Kit A07 | 50 $\mu$L | 600 $\mu$L | 650 $\mu$L | 13 |
| Kit A08 | 50 $\mu$L | 400 $\mu$L | 450 $\mu$L | 9 |
| Kit A09 | 50 $\mu$L | 500 $\mu$L | 550 $\mu$L | 11 |
| Kit A10 | 50 $\mu$L | 300 $\mu$L | 350 $\mu$L | 7 |
| Kit A11 | 50 $\mu$L | 400 $\mu$L | 450 $\mu$L | 9 |
| Kit A12 | 50 $\mu$L | 500 $\mu$L | 550 $\mu$L | 11 |
| Kit A13 | 50 $\mu$L | 300 $\mu$L | 350 $\mu$L | 7 |
| Kit A14 | 50 $\mu$L | 280 $\mu$L | 330 $\mu$L | 6.6 |
| Kit A15 | 50 $\mu$L | 500 $\mu$L | 550 $\mu$L | 11 |
| Kit A16 | 50 $\mu$L | 350 $\mu$L | 400 $\mu$L | 8 |
| Kit A17 | 50 $\mu$L | 500 $\mu$L | 550 $\mu$L | 11 |
| Kit A18 | 50 $\mu$L | 400 $\mu$L | 450 $\mu$L | 9 |
| Kit A19 | 50 $\mu$L | 300 $\mu$L | 350 $\mu$L | 7 |

[a]The volumes of extraction solution were obtained from the instructions or reagent manufacturers and verified using manual pipettes.

detection ranged between $1.28 \times 10^2$ to $1.02 \times 10^3$, $1.28 \times 10^2$ to $4.10 \times 10^3$, and $1.28 \times 10^2$ to $2.05 \times 10^3$ copies/mL for the wild-type (WT), Delta, and Omicron strains, respectively, corresponding to $7.81 \times 10^3$ to $6.25 \times 10^4$, $7.81 \times 10^3$ to $2.50 \times 10^5$, and $7.81 \times 10^3$ to $1.25 \times 10^5$ copies/mL of the 50 $\mu$L original sample. For the 20-in-1 pooling tests and 5 rRT-PCR detection kits, the lowest concentrations ranged between $1.30 \times 10^2$ to $1.04 \times 10^3$, $5.19 \times 10^2$ to $2.07 \times 10^3$, and $2.59 \times 10^2$ to $1.04 \times 10^3$ copies/mL for the WT, Delta, and Omicron strains, respectively. Kit N03 reliably detected as few as $10^2$ copies/mL for both sampling strategies, showing the most sensitive performance compared with the other rRT-PCR kits. Unexpectedly, the POCT Kit P01 showed a comparable analytical sensitivity to those of the rRT-PCR kits with the lowest concentration of $1.42 \times 10^3$ copies/mL for all three strains (Table 1; Supplemental Material File 3).

Overall, when detecting the diluted samples higher than the claimed limits of detection (LODs), the 20-in-1 pooling tests could substantially achieve the same detection performance as could the single sampling tests. When it came to the Delta (22/27 versus 15/27, $P = 0.040$) and Omicron (22/27 versus 14/27, $P = 0.021$) variants below than the claimed LODs, impaired analytical sensitivity was found in the 20-in-1 pooling strategy using Kit N03. In terms of test results among variants, in contrast to the WT strain, the Delta variant adversely affected the analytical sensitivity of Kit N04, both for the single sampling tests (19/27 versus 11/27, $P = 0.028$) and for the 20-in-1 pooling tests (16/27 versus 7/27, $P = 0.013$). Also, the Omicron variant had no significant effect on the 5 rRT-PCR kits and the POCT kit.

**Rapid antigen tests.** Regarding the testing of the inactivated cell culture supernatants, the lowest virus concentrations to achieve a 100% detection rate ranged between $2.84 \times 10^5$ to $7.14 \times 10^6$, $8.68 \times 10^4$ to $7.14 \times 10^6$, and $1.12 \times 10^5$ to $3.57 \times 10^6$ copies/mL for the WT, Delta, and Omicron strains, respectively, corresponding to $3.12 \times 10^6$ to

$5.00 \times 10^7$, $7.81 \times 10^5$ to $5.00 \times 10^7$, and $7.81 \times 10^5$ to $2.50 \times 10^7$ copies/mL of the 50 $\mu$L original sample (Table 2; Supplemental Material File 3). We also tested 19 rapid antigen tests with 5 recombinant SARS-CoV-2 N proteins. The lowest concentrations ranged from 3.47 to 142.86 ng/mL, 1.74 to 142.86 ng/mL, 3.47 to 142.86 ng/mL, 3.47 to 142.86 ng/mL, and 5.68 to 142.86 ng/mL for the WT, Delta AY.2, Delta AY.1/AY.3, Omicron BA.1, and Omicron BA.2 recombinant N proteins, respectively (Table 3; Supplemental Material File 3). Almost all of the rapid antigen tests reliably detected around $2.50 \times 10^7$ copies/mL of 50 $\mu$L of inactivated supernatants and 500 ng/mL of 50 $\mu$L of recombinant N proteins.

In contrast to the rRT-PCR kits, great variations in analytical sensitivity were observed among the rapid antigen kits. The lowest concentrations that achieved a 100% rate of detection success ranged from $10^4$ to $10^6$ copies/mL for inactivated cell culture supernatants and from 1 to 150 ng/mL for recombinant N proteins. The best analytical performance was achieved by Kit A05 in detecting inactivated viruses. For detecting recombinant N proteins, Kit A01 performed the best. The assay manufactured by Kit A19 was considerably less sensitive than the other assays in detecting both inactivated viruses and recombinant N proteins. However, for each rapid antigen test, no significant differences in analytical sensitivity were found among the 3 inactivated cell culture supernatants or the 5 recombinant N proteins.

## DISCUSSION

In China, since May of 2021, the Delta variant with an $R_0$ value below 7 (original strain 2.5) has ravaged many cities, including Guangzhou, Nanjing, Yangzhou, Putian, Xiamen, and Ejina Banner. The Omicron variant is estimated to have an $R_0$ value of up to 10 and a doubling time of every 2 to 3 days (18), which makes it reasonable to supersede Delta as the dominant variant by December of 2021. With the proportion of asymptomatic infections calculated to be as high as 80 to 90%, as well as the rapid occult transmissibility (19), the Omicron variant has swept numerous cities with an unprecedented speed, especially in Shanghai, where the number of daily confirmed infected cases has roared up to 20,000 for days on end. Faced with the grim situation of the SARS-CoV-2 pandemic and the Omicron variant, which has a higher pathogenicity and spreads more rapidly, frequent testing enables the early detection of infections, which is critical in providing a prompt diagnosis for patient management and in conducting epidemiological studies to promote public health measures (20–22). Several studies (23, 24) have revealed that nearly one-fifth of all virus transmission was recognized to be associated with asymptomatic or presymptomatic individuals, and the efficacy of outbreak control depends mostly on the frequency of testing, rather than on test sensitivity, through epidemiological modeling. Therefore, pandemic prevention and control strategies are required to cut off the transmission chains within communities through the scale-up of diagnostic testing, contact tracing, and quarantining (25).

The continuous COVID-19 pandemic has promoted the occurrence and development of a variety of diagnostic strategies based on different principles. Due to the urgent demand during the unremitting transmission and evolution of SARS-CoV-2, the performance characteristic of analytical sensitivity has not been thoroughly demonstrated for the different detection methods. In this study, a 50 $\mu$L uniform volume of WT, Delta, and Omicron cell culture supernatants with given concentrations were used to obtain approximate ranges of the analytical sensitivities of nucleic acid amplification tests (NAATs; single sampling tests, 20-in-1 pooling tests, and nucleic acid POCT) and rapid antigen tests applied in different scenarios.

At the beginning of the outbreak, the single sampling test of rRT-PCR was regarded as the most sensitive and specific method for the detection of SARS-CoV-2, and it was recommended for confirming infected cases and testing specific groups, including the contacts of confirmed or frequently exposed groups (26), to ensure the timely implementation of public health measures and patient management procedures, such as contact tracing and quarantine. In the study, for the single sampling tests, the lowest inactivated virus concentrations at 100% detection rates for the five detection kits most commonly used in China were found to have a range of around $1 \times 10^2$ and $5 \times 10^3$ copies/mL, with Kit N03 performing the best. Large-scale population screening via the rRT-PCR method plays a crucial

**TABLE 2** *In vitro* analytical sensitivity of 4 detection methods to 3 inactivated cell culture supernatants

| Detection kits | Wild-type strain (copies/mL) | | Delta variant (copies/mL) | | Omicron variant (copies/mL) | |
|---|---|---|---|---|---|---|
| | Lowest concentration at 100% detection rates[a] | Concentration equivalent to the 50 μL sample[b] | Lowest concentration at 100% detection rates[a] | Concentration equivalent to the 50 μL sample[b] | Lowest concentration at 100% detection rates[a] | Concentration equivalent to the 50 μL sample[b] |
| **Single sampling tests** | | | | | | |
| Kit N01 | $1.02 \times 10^3$ | $6.25 \times 10^4$ | $2.05 \times 10^3$ | $1.25 \times 10^5$ | $2.05 \times 10^3$ | $1.25 \times 10^5$ |
| Kit N02 | $1.02 \times 10^3$ | $6.25 \times 10^4$ | $2.05 \times 10^3$ | $1.25 \times 10^5$ | $1.02 \times 10^3$ | $6.25 \times 10^4$ |
| Kit N03 | $1.28 \times 10^2$ | $7.81 \times 10^3$ | $1.28 \times 10^2$ | $7.81 \times 10^3$ | $1.28 \times 10^2$ | $7.81 \times 10^3$ |
| Kit N04 | $5.12 \times 10^2$ | $3.12 \times 10^4$ | $2.05 \times 10^3$ | $1.25 \times 10^5$ | $1.02 \times 10^3$ | $6.25 \times 10^4$ |
| Kit N05 | $1.02 \times 10^3$ | $6.25 \times 10^4$ | $4.10 \times 10^3$ | $2.50 \times 10^5$ | $1.02 \times 10^3$ | $6.25 \times 10^4$ |
| **20-in-1 pooling tests** | | | | | | |
| Kit N01 | $1.04 \times 10^3$ | $2.50 \times 10^5$ | $2.07 \times 10^3$ | $5.00 \times 10^5$ | $1.04 \times 10^3$ | $2.50 \times 10^5$ |
| Kit N02 | $1.04 \times 10^3$ | $2.50 \times 10^5$ | $1.04 \times 10^3$ | $2.50 \times 10^5$ | $1.04 \times 10^3$ | $2.50 \times 10^5$ |
| Kit N03 | $1.30 \times 10^2$ | $3.12 \times 10^4$ | $5.19 \times 10^2$ | $1.25 \times 10^5$ | $2.59 \times 10^2$ | $6.25 \times 10^4$ |
| Kit N04 | $2.59 \times 10^2$ | $6.25 \times 10^4$ | $1.04 \times 10^3$ | $2.50 \times 10^5$ | $5.19 \times 10^2$ | $1.25 \times 10^5$ |
| Kit N05 | $5.19 \times 10^2$ | $1.25 \times 10^5$ | $2.07 \times 10^3$ | $5.00 \times 10^5$ | $1.04 \times 10^3$ | $2.50 \times 10^5$ |
| **Point-of-care test** | | | | | | |
| Kit P01 | $1.42 \times 10^3$ | $1.56 \times 10^4$ | $1.42 \times 10^3$ | $1.56 \times 10^4$ | $1.42 \times 10^3$ | $1.56 \times 10^4$ |
| **Rapid antigen tests** | | | | | | |
| Kit A01 | $3.47 \times 10^5$ | $3.12 \times 10^6$ | $8.68 \times 10^4$ | $7.81 \times 10^5$ | $1.74 \times 10^5$ | $1.56 \times 10^6$ |
| Kit A02 | $2.84 \times 10^5$ | $3.12 \times 10^6$ | $1.42 \times 10^5$ | $1.56 \times 10^6$ | $1.42 \times 10^5$ | $1.56 \times 10^6$ |
| Kit A03 | $1.14 \times 10^6$ | $1.25 \times 10^7$ | $5.68 \times 10^5$ | $6.25 \times 10^6$ | $5.68 \times 10^5$ | $6.25 \times 10^6$ |
| Kit A04 | $4.01 \times 10^5$ | $3.12 \times 10^6$ | $2.00 \times 10^5$ | $1.56 \times 10^6$ | $2.00 \times 10^5$ | $1.56 \times 10^6$ |
| Kit A05 | $4.46 \times 10^5$ | $3.12 \times 10^6$ | $1.12 \times 10^5$ | $7.81 \times 10^5$ | $1.12 \times 10^5$ | $7.81 \times 10^5$ |
| Kit A06 | $3.57 \times 10^6$ | $2.50 \times 10^7$ | $1.79 \times 10^6$ | $1.25 \times 10^7$ | $1.79 \times 10^6$ | $6.25 \times 10^6$ |
| Kit A07 | $1.92 \times 10^6$ | $2.50 \times 10^7$ | $9.62 \times 10^5$ | $1.25 \times 10^7$ | $4.81 \times 10^5$ | $1.25 \times 10^7$ |
| Kit A08 | $2.78 \times 10^6$ | $2.50 \times 10^7$ | $6.94 \times 10^5$ | $6.25 \times 10^6$ | $1.39 \times 10^6$ | $1.25 \times 10^7$ |
| Kit A09 | $1.14 \times 10^6$ | $1.25 \times 10^7$ | $5.68 \times 10^5$ | $6.25 \times 10^6$ | $5.68 \times 10^5$ | $6.25 \times 10^6$ |
| Kit A10 | $1.79 \times 10^6$ | $1.25 \times 10^7$ | $8.93 \times 10^5$ | $6.25 \times 10^6$ | $8.93 \times 10^5$ | $6.25 \times 10^6$ |
| Kit A11 | $2.78 \times 10^6$ | $2.50 \times 10^7$ | $1.39 \times 10^6$ | $1.25 \times 10^7$ | $1.39 \times 10^6$ | $1.25 \times 10^7$ |
| Kit A12 | $2.27 \times 10^6$ | $2.50 \times 10^7$ | $1.14 \times 10^6$ | $1.25 \times 10^7$ | $1.14 \times 10^6$ | $1.25 \times 10^7$ |
| Kit A13 | $4.46 \times 10^5$ | $3.12 \times 10^6$ | $4.46 \times 10^5$ | $3.12 \times 10^6$ | $2.23 \times 10^5$ | $1.56 \times 10^6$ |
| Kit A14 | $1.89 \times 10^6$ | $1.25 \times 10^7$ | $1.89 \times 10^6$ | $1.25 \times 10^7$ | $1.89 \times 10^6$ | $1.25 \times 10^7$ |
| Kit A15 | $1.14 \times 10^6$ | $1.25 \times 10^7$ | $5.68 \times 10^5$ | $6.25 \times 10^6$ | $1.14 \times 10^6$ | $1.25 \times 10^7$ |
| Kit A16 | $3.12 \times 10^6$ | $2.50 \times 10^7$ | $1.56 \times 10^6$ | $1.25 \times 10^7$ | $3.12 \times 10^6$ | $2.50 \times 10^7$ |
| Kit A17 | $5.68 \times 10^5$ | $6.25 \times 10^6$ | $2.84 \times 10^5$ | $3.12 \times 10^6$ | $2.84 \times 10^5$ | $3.12 \times 10^6$ |
| Kit A18 | $1.39 \times 10^6$ | $1.25 \times 10^7$ | $1.39 \times 10^6$ | $1.25 \times 10^7$ | $1.39 \times 10^6$ | $1.25 \times 10^7$ |
| Kit A19 | $7.14 \times 10^6$ | $5.00 \times 10^7$ | $7.14 \times 10^6$ | $5.00 \times 10^7$ | $3.57 \times 10^6$ | $2.50 \times 10^7$ |

[a]Given the different volumes of sample preservation solution or sample extraction solution, the lowest final concentrations at 100% detection rates were calculated.

[b]Concentration of the 50 μL original samples before being added to the sample preservation solution or sample extraction solution.

TABLE 3 *In vitro* analytical sensitivity of 19 rapid antigen tests to 5 recombinant N proteins

| Detection kits | Wild-type strain (ng/mL) | | Delta AY.2 sublineage (ng/mL) | | Delta AY.1/AY.3 sublineage (ng/mL) | | Omicron BA.1 sublineage (ng/mL) | | Omicron BA.2 sublineage (ng/mL) | |
|---|---|---|---|---|---|---|---|---|---|---|
| | Lowest concentration at 100% detection rate[a] | Concentration equivalent to the 50 $\mu$L sample[b] | Lowest concentration at 100% detection rate[a] | Concentration equivalent to the 50 $\mu$L sample[b] | Lowest concentration at 100% detection rate[a] | Concentration equivalent to the 50 $\mu$L sample[b] | Lowest concentration at 100% detection rate[a] | Concentration equivalent to the 50 $\mu$L sample[b] | Lowest concentration at 100% detection rate[a] | Concentration equivalent to the 50 $\mu$L sample[b] |
| Kit A01 | 3.47 | 31.25 | 1.74 | 15.62 | 3.47 | 31.25 | 3.47 | 31.25 | 6.94 | 62.5 |
| Kit A02 | 5.68 | 62.5 | 2.84 | 31.25 | 5.68 | 62.5 | 5.68 | 62.5 | 5.68 | 62.5 |
| Kit A03 | 22.73 | 250 | 22.73 | 250 | 45.45 | 500 | 22.73 | 250 | 22.73 | 250 |
| Kit A04 | 4.01 | 31.25 | 4.01 | 31.25 | 4.01 | 31.25 | 4.01 | 31.25 | 8.01 | 62.5 |
| Kit A05 | 35.71 | 250 | 8.93 | 62.5 | 35.71 | 250 | 17.86 | 125 | 35.71 | 250 |
| Kit A06 | 35.71 | 250 | 17.86 | 125 | 35.71 | 250 | 35.71 | 250 | 35.71 | 250 |
| Kit A07 | 38.46 | 500 | 19.23 | 250 | 19.23 | 250 | 19.23 | 250 | 19.23 | 250 |
| Kit A08 | 27.78 | 250 | 13.89 | 125 | 27.78 | 250 | 13.89 | 125 | 13.89 | 125 |
| Kit A09 | 22.73 | 250 | 11.36 | 125 | 45.45 | 500 | 22.73 | 250 | 22.73 | 250 |
| Kit A10 | 35.71 | 250 | 8.93 | 62.5 | 8.93 | 62.5 | 71.43 | 500 | 71.43 | 500 |
| Kit A11 | 27.78 | 250 | 27.78 | 250 | 27.78 | 250 | 27.78 | 250 | 27.78 | 250 |
| Kit A12 | 22.73 | 250 | 11.36 | 125 | 22.73 | 250 | 22.73 | 250 | 22.73 | 250 |
| Kit A13 | 8.93 | 62.5 | 4.46 | 31.25 | 17.86 | 125 | 8.93 | 62.5 | 8.93 | 62.5 |
| Kit A14 | 18.94 | 125 | 9.47 | 62.5 | 18.94 | 125 | 18.94 | 125 | 18.94 | 125 |
| Kit A15 | 45.45 | 500 | 45.45 | 500 | 11.36 | 125 | 45.45 | 500 | 45.45 | 500 |
| Kit A16 | 62.50 | 500 | 15.63 | 125 | 62.50 | 500 | 15.63 | 125 | 62.50 | 500 |
| Kit A17 | 5.68 | 62.5 | 2.84 | 31.25 | 5.68 | 62.5 | 5.68 | 62.5 | 5.68 | 62.5 |
| Kit A18 | 55.56 | 500 | 111.11 | 1,000 | 111.11 | 1,000 | 111.11 | 1,000 | 111.11 | 1,000 |
| Kit A19 | 142.86 | 1,000 | 142.86 | 1,000 | 142.86 | 1,000 | 142.86 | 1,000 | 142.86 | 1,000 |

[a]Given the different volumes of sample preservation solution or sample extraction solution, the lowest final concentrations at 100% detection rates were calculated.
[b]Concentration of the 50 $\mu$L original sample before being added to the sample preservation solution or sample extraction solution.

role in the identification of COVID-19 positive cases. In the quest for improving overall detection efficiency with an acceptable slight loss of analytical sensitivity, the pooling strategies are proposed, and these can be implemented to expedite the early discovery of community transmission, promote timely infection control measures of SARS-CoV-2, and alleviate the workload of medical staff. By mixing nasopharyngeal and oropharyngeal swabs and testing them as a single pool, the detection kits and turnaround times are largely economized. Catherine et al. (27) showed that a 9/10-in-1 pooling strategy could detect positive samples correctly at a population prevalence rate of 0.07%. Idan et al. (28) found that the false-negative rate of a 32-in-1 pooling strategy can be as low as 10%. Stefan et al. (14) proposed a 30-in-1 pooling strategy for increasing test efficiency on the premise of ensuring the correct detection of positive samples. However, the primary bottleneck of pooled testing is the reduced concentration of viral genetic material below the limit of detection for certain tests due to sample dilution, which thereby leads to decreased diagnostic sensitivity and false-negative results (29). When using the 5 rRT-PCR kits to detect the same strain, the range of the lowest inactivated virus concentration to achieve 100% detection via 20-in-1 pooling tests was substantially the same as that of the single sampling tests. Overall, when detecting the diluted samples above the claimed LODs, the 20-in-1 pooling tests showed comparable analytical sensitivities to those of the single sampling tests. Studies (30) have found that the viral load at the early onset was beyond $1\times10^6$ copies/mL, which can be detected by both single sampling tests and 20-in-1 pooling tests. Therefore, the 20-in-1 pooling tests enable mass nucleic acid testing with only 5% of the original testing workload, greatly improving the daily testing efficiency, which is the first consideration in the screening of asymptomatic infected cases in low-risk regions (31). Nevertheless, regarding the possibility of the increasing need to retest singly, the optimal pool size, while ensuring the expected analytical sensitivity and time efficiency, requires the deliberation of the prevalence rates in the community, the infection situations, the aim of the testing, and the available resources. In addition, due to the Omicron variant's capability of aerosol transmission (32), prevention measures, including mask use, well-zoned sampling sites, and sufficiently dispersed staff density should be strictly implemented during the organization of sample collection.

The nucleic acid point-of-care test has the advantage of not relying on laboratory equipment and professional personnel to detect viral nucleic acid, but its extensive deployment in domestic regions is impeded by the development and manufacturing of technology platforms. Furthermore, the sensitivity level of self-administered nucleic acid POCT for the detection of SARS-CoV-2 RNA also varies, depending on the methods and application settings. Unexpectedly, the Ustar nucleic acid POCT, which combines cross-priming amplification (CPA) and nucleic acid lateral flow technologies, can detect samples at concentrations as low as $1.42 \times 10^3$ copies/mL, showing excellent performance in the detection of SARS-CoV-2 in the study. With a short turnaround time, portable procedures (33), and an analytical sensitivity comparable to that of the gold standard rRT-PCR, nucleic acid POCT is a promising diagnostic agent for aiding in the expansion of testing (34) and can be used as confirmatory testing where laboratory-based nucleic acid amplification testing is not available (35). However, due to the scarcity of portable nucleic acid POCT assays validated in China, only one CE-IVD marked kit was included in the study. More evidence-based experimental data remains to be explored in order to obtain a comprehensive view of the performance of this method in detecting SARS-CoV-2.

Rapid antigen tests are developing rapidly and are becoming available from various manufacturers in response to the Omicron variant wreaking havoc in many regions of China. With the advantages of being free of laboratory settings, offering a shorter time to result, and being mass and cost-effective to manufacture, they can act as a potential alternative method if molecular tests are not available (36). Our study revealed that the detection performance of different rapid antigen kits varied greatly, and the lowest concentrations that achieved a 100% rate of detection success ranged from $10^4$ to $10^6$ copies/mL for inactivated cell culture supernatants and from 1 to 150 ng/mL for recombinant N proteins, which was essentially consistent with the findings reported in

previously published articles (35, 37). The antibody labeling method of rapid antigen tests would affect their analytical performance, and the fluorescence microsphere was observed to enhance the sensitivity of the analytical signal by 10 to 100-fold compared to that of the latex microsphere and colloidal gold (38). Besides, the lowest concentrations of the 50 $\mu$L original sample that displayed 100% detection by rapid antigen tests ($10^5$ to $10^7$ copies/mL) were about 100 times as high as those of the single sampling test ($10^3$ to $10^5$ copies/mL). On account of the viral load changing rapidly during the acute phase of infection, rapid antigen tests can only be applied validly in the first week of symptoms (39). Studies (40–42) have also found that there is little chance of transmitting the virus for patients with a viral load of less than $1 \times 10^6$ copies/mL at the end of the first week of symptoms, meaning that rapid antigen tests can be used as tools by which to estimate infectivity and that a negative antigen result is an indication of infectivity resolution. On the whole, although they are not as sensitive and specific as molecular tests for the diagnosis of SARS-CoV-2 infection, rapid antigen tests should be considered as rapid diagnostic tools that require minimal training and offer cost-effective and time-saving detection processes for the instantaneous assessment of infectivity rather than for the exclusion of infection, mass screening in high-risk areas, and the detection of symptomatic cases meeting the COVID-19 case definition (43). However, a randomized clinical trial (44) found that a large proportion of the population misinterpreted the negative results of at-home self-tests or ignored the CDC's recommendations to self-isolate, creating redundant disruptions and unexpected risks. Consequently, user compliance and the interpretability of the results should be improved to maximize the benefits of at-home self-test kits.

The indispensable factors for a correct diagnosis of SARS-CoV-2 include the attributes of the detection method itself, the sampling time after the onset of symptoms, the quality of the specimen, the proficiency of the test, and the interpretation of the result. Also, the continuous evolution of SARS-CoV-2 affects the accuracy and reliability of detection results to a certain extent. Once the mutations of the virus are located at the primer/probe-targeted regions, the effectiveness of molecular assays might be influenced to some degree. In this study, in contrast to the WT strain, impaired analytical sensitivity was found in the Delta variant using Kit N04, both for the single sampling tests (19/27 versus 11/27, $P = 0.028$) and for the 20-in-1 pooling tests (16/27 versus 7/27, $P = 0.013$), whereas the Omicron variant had no significant effect on the 5 rRT-PCR kits and the POCT kit, which is in concordance with the findings of a prior study (45). However, the 19 rapid antigen tests showed an insignificant difference in analytical sensitivity for the 3 inactivated cell culture supernatants and the 5 recombinant N proteins, which might be attributable to the fact that nearly all of the rapid antigen tests target the SARS-CoV-2 nucleocapsid protein, which has less of a probability to mutate (12). Hence, to ensure the validity of the different detection strategies under the background of SARS-CoV-2 genomic diversity, it is essential to consolidate genomic surveillance and track the potential effects of mutations on detection performance.

In conclusion, the study evaluated the analytical sensitivity of 4 SARS-CoV-2 detection strategies applied in different settings and provided helpful insights into the scientific deployment of these tests (Table 4). Generally, the analytical sensitivities of nucleic acid amplification tests were superior to those of rapid antigen tests, with the single sampling strategy showing the highest. In a specific scenario, the optimal strategy should be adopted in consideration of the testing purpose, resource availability, cost performance, and result rapidity on the premise of test accuracy to thereby improve the overall detection efficiency, facilitate the discovery of early community transmission, and enable timely and long-term infection control measures under the conditions of limited detection capacity and overburdened laboratory infrastructure.

## MATERIALS AND METHODS

**Study design and samples.** We simulated the application scenarios of 4 strategies for COVID-19 detection, including NAATs (single sampling testing, 20-in-1 pooling testing, and point-of-care testing) and rapid antigen testing. The single-center evaluation of analytical sensitivity was based on SARS-CoV-2 cell culture

**TABLE 4** Characteristics and recommended application scenarios of 4 detection strategies for SARS-CoV-2

| Detection strategy | Advantages | Disadvantages | Application scenarios |
|---|---|---|---|
| Single sampling test | Most sensitive and specific | High cost<br>Limited throughput<br>Long turnaround time<br>Require sophisticated equipment and professional personnel | Confirm infected patients<br>Detect specific groups, including contacts of confirmed or frequently exposed groups |
| 20-in-1 pooling test | Relatively sensitive<br>Improved daily detection efficiency | Long turnaround time<br>Require sophisticated equipment and professional personnel | Large-scale screening in low-risk regions<br>Early detection of community transmission |
| Nucleic acid Ppoint-of-care test | Relatively sensitive<br>Simple operation<br>Short turnaround time<br>Independent of laboratory settings | Limited by the development and manufacturing of the technology platform<br>Poor user compliance | Aid testing expansion<br>Confirmatory testing where laboratory-based nucleic acid amplification testing is not available |
| Rapid antigen test | Less expensive and easier to operate<br>Fast result rapidity<br>Independent of laboratory settings | Less sensitive<br>Uncertain of detection quality<br>Poor user compliance | Mass screening in high-risk regions<br>Instantaneous assessment of infectiousness<br>Detect symptomatic cases meeting the COVID-19 case definition<br>Alternative method if molecular tests are not available |

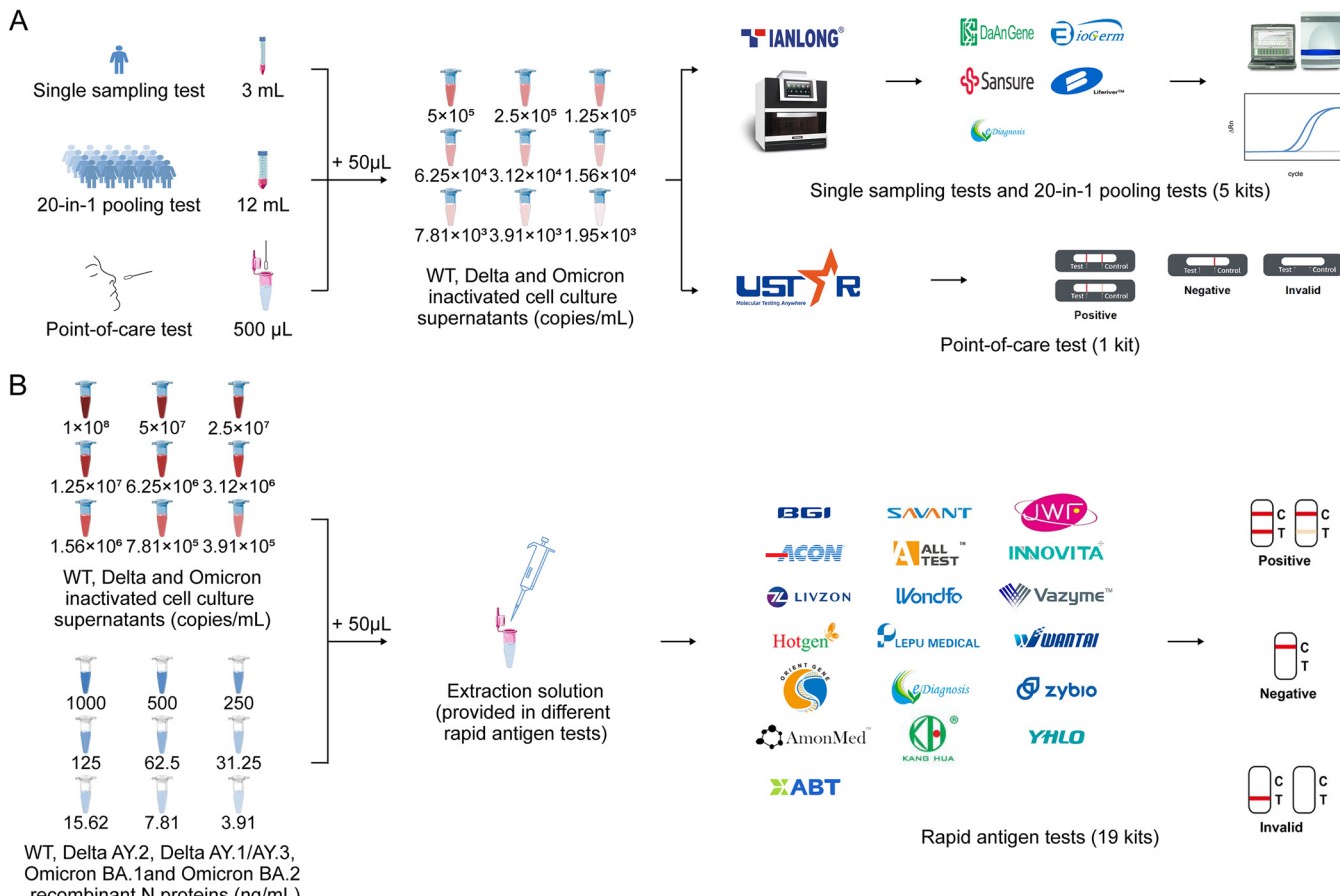

**FIG 1** Schematic diagram of the study design. (A) Nucleic acid amplification tests, including single sampling tests, 20-in-1 pooling tests, and nucleic acid point-of-care tests (POCT). (B) Rapid antigen tests.

supernatants with determined viral loads. As a complement, recombinant SARS-CoV-2 N proteins were also used for the rapid antigen tests (Fig. 1).

3 inactivated cell culture supernatants of SARS-CoV-2 WT, Delta, and Omicron strains were provided by the Sinovac Biotech Co., Ltd. (China). For quantification, viral RNAs with proper dilutions were extracted using a QIAamp Viral RNA Minikit (Qiagen, Hilden, Germany) and were reverse transcribed into cDNAs using a PrimeScript RT Reagent Kit (Perfect Real Time; TaKaRa, Japan). Subsequently, the samples were subjected to droplet digital PCR (ddPCR) on a Bio-Rad QX-200 System (USA), using the N gene assay recommended by China's Centers for Disease Control (CDC). 5 recombinant SARS-CoV-2 N proteins of the SARS-CoV-2 WT strain (cat. number 40588-V07E), Delta AY.2 sublineage (cat. number 40588-V07E29), Delta AY.1/AY.3 sublineage (cat number 40588-V07E32), Omicron BA.1 sublineage (cat number 40588-V07E34), and Omicron BA.2 sublineage (cat number 40588-V07E35) were obtained from Sino Biological, Inc. (China). The concentrations of the recombinant N proteins were measured using a Qubit 3.0 fluorometer with the Qubit Protein Assay Kit (Thermo Fisher Scientific, USA).

**Nucleic acid amplification tests.** To evaluate the analytical sensitivity in real-world sampling situations, a total of 1 and 20 oropharyngeal swabs from SARS-CoV-2-negative healthy volunteers were placed into a single sampling tube containing 3 mL of a sample preservation solution and a 50 mL centrifuge tube containing 12 mL of a sample preservation solution, respectively. Each sample was prepared 27 times. After full oscillation and blending, serial 2-fold dilutions of the WT, Delta, and Omicron cell culture supernatants were performed ($5 \times 10^5$, $2.5 \times 10^5$, $1.25 \times 10^5$, $6.25 \times 10^4$, $3.12 \times 10^4$, $1.56 \times 10^4$, $7.81 \times 10^3$, $3.91 \times 10^3$, and $1.95 \times 10^3$ copies/mL), and 50 $\mu$L of them were added to each sample, respectively. Remarkably, this step with positive supernatants diluted in sample preservation solution introduces a predilution effect (1:61 as 50 $\mu$L in 3,050 $\mu$L for the single sampling test and 1:241 as 50 $\mu$L in 12,050 $\mu$L for the 20-in-1 pooling test), leading to a loss of sensitivity as claimed by the reagent manufacturers. This factor was staken into account when calculating the analytical sensitivity afterwards. Nucleic acid was extracted using a Tianlong automatic nucleic acid extraction system (NP968-C, Xi'an Tianlong Science and Technology Co., Ltd.), and tested in triplicate using 5 NMPA approved SARS-CoV-2 rRT-PCR kits (Kit N01, Daan Gene Co., Ltd. of Sun Yat-sen University; Kit N02, Shanghai BioGerm Medical Technology Co., Ltd.; Kit N03, Sansure Bio-tech Co., Ltd.; Kit N04, Shanghai Liferiver BioTech Co., Ltd.; and Kit N05, Wuhan EasyDiagnosis Biomedicine Co., Ltd.) (Table 5). The retesting and interpretation of results were performed according to the manufacturers' instructions (Supplemental Material File 1; Table S1).

**TABLE 5** Characteristics of SARS-CoV-2 nucleic acid amplification tests and rapid antigen tests[a]

| Detection kit | Code[b] | Specimens | Detection target | Claimed limit of detection | RNA input (μL/sample input) | Total reaction volume (μL) | PCR cycle number |
|---|---|---|---|---|---|---|---|
| **Real-time RT-PCR tests** | | | | | | | |
| Daan | Kit N01 | NP, OP, sputum | ORF1ab, N genes | 500 copies/mL | 5 | 25 | 45 |
| BioGerm | Kit N02 | OP, sputum | ORF1ab, N genes | 500 copies/mL | 5 | 25 | 45 |
| Sansure | Kit N03 | NP, OP, BALF | ORF1ab, N genes | 200 copies/mL | 20 | 50 | 45 |
| Liferiver | Kit N04 | NP, sputum | ORF1ab, N, E genes | 200 copies/mL | 5 | 25 | 45 |
| EasyDiagnosis | Kit N05 | NP, OP, sputum | ORF1ab, N genes | 500 copies/mL | 5 | 25 | 40 |
| **Point-of-care test** | | | | | | | |
| Ustar | Kit P01 | Anterior nasal swab | ORF1ab gene | 3,000 copies/mL | 30 μL/1 drop | /[c] | / |
| **Rapid antigen tests** | | | | | | | |
| BGI | Kit A01 | NP, OP | N protein | 5 pg/mL, $1 \times 10^4$ copies/mL, 150 $TCID_{50}$/mL | 80 μL | / | / |
| Savant | Kit A02 | NP, OP | N protein | / | 90 μL/3 to 4 drops | / | / |
| Jinwofu | Kit A03 | NP, OP, nasal swab | N protein | 100 $TCID_{50}$/mL | 2 to 3 drops | / | / |
| ACON | Kit A04 | Anterior nasal swab | N protein | 160 $TCID_{50}$/mL | 4 drops | / | / |
| AllTest | Kit A05 | Nasal swab | N, S proteins | 78 $TCID_{50}$/mL | 75 to 100 μL/3 to 4 drops | / | / |
| Innovita | Kit A06 | NP, nasal swab | N protein | 125 $TCID_{50}$/mL | 80 μL/3 drops | / | / |
| Livzon | Kit A07 | NP, OP, nasal swab | N protein | 100 $TCID_{50}$/mL | 100 μL/4 drops | / | / |
| Wondfo | Kit A08 | NP, OP | N protein | 850 $TCID_{50}$/mL | 80 μL/3 to 4 drops | / | / |
| Vazyme | Kit A09 | NP, OP, nasal swab | N, S proteins | 50 $TCID_{50}$/mL | 80 μL/4 drops | / | / |
| Hotgen | Kit A10 | Nasal swab | N protein | / | 100 μL/4 drops | / | / |
| Lepu | Kit A11 | Nasal swab | N protein | 200 $TCID_{50}$/mL | 100 μL/3 drops | / | / |
| Wantai | Kit A12 | NP, OP, nasal swab | N protein | 137 $TCID_{50}$/mL | 80 μL/4 drops | / | / |
| Orient Gene | Kit A13 | Nasal swab | / | / | 4 drops | / | / |
| EasyDiagnosis | Kit A14 | NP, OP, nasal swab | N protein | 500 $TCID_{50}$/mL | 120 μL/3 drops | / | / |
| Zybio | Kit A15 | NP, OP, nasal swab | N protein | 70 $TCID_{50}$/mL | 75 μL/4 drops | / | / |
| AmonMed | Kit A16 | NP, OP, nasal swab | N protein | 600 $TCID_{50}$/mL | 2 drops | / | / |
| Kanghua | Kit A17 | OP, nasal swab | N protein | 64 $TCID_{50}$/mL | 60 to 80 μL/2 to 3 drops | / | / |
| YHLO | Kit A18 | NP, Nasal swab | N protein | 250 $TCID_{50}$/mL | 100 μL/3 drops | / | / |
| XABT | Kit A19 | NP, Nasal swab | N protein | 200 $TCID_{50}$/mL | 75 to 100 μL/3 to 4 drops | / | / |

[a]NP, nasal pharyngeal; OP, oral pharyngeal; BALF, bronchoalveolar lavage fluid; ORF, open reading frame; N, nucleocapsid protein gene; S, spike protein; $TCID_{50}$, median tissue culture infectious dose; RT-PCR, reverse transcriptase polymerase chain reaction.

[b]For the ease of communication throughout the article, each real-time a RT-PCR kit was assigned a code from Kit N01 to Kit N05, point-of-care testing was encoded as Kit P01, and each rapid antigen test was assigned a code from Kit A01 to Kit A19.

[c]/, Items were unavailable from the instructions of detection kits.

Nucleic acid POCTs are portable, easy-to-operate, and isothermal amplification-based devices that are mainly characterized by their relative sensitivity, simple operation, short turnaround time, and independence of laboratory settings (33). The EasyNAT COVID-19 RNA Test from Ustar Biotechnologies Ltd. carries out CPA reactions through specific primers, probes, and DNA polymerase with high strand displacement activity to qualitatively detect the ORF1ab gene of SARS-CoV-2 with a claimed LOD of $3 \times 10^3$ copies/mL (46). Self-collected nasopharyngeal swabs are immersed in the lysis buffer, one drop of which is added to a module preloaded with nucleic acid amplification reagents. After 55 min of reaction, the results can be available in the corresponding lateral flow strips. To identify the analytical sensitivity of the handheld portable POCT product (coded as Kit P01), 50 $\mu$L of the above supernatants were introduced into the lysis buffer provided with the kit in triplicate, which resulted in a predilution effect of 1:11. The subsequent procedures and interpretation of results were carried out as per the instructions.

**Rapid antigen tests.** The main principle behind rapid antigen testing is the use of lateral flow immunoassays designed with the fluorescence microsphere, latex microsphere, or colloidal gold labeled SARS-CoV-2 protein antibody to form an antibody-antigen (Ab-Ag) complex. Within 15 min, the test results can be interpreted with simple instruments when using the fluorescence method or with the naked eye when using the latex and colloidal gold methods. We included 19 rapid antigen tests approved by the NMPA in our study, including two fluorescence immunochromatography methods (Kit A01, BGI Biotech Co., Ltd. and Kit A02, Beijing Savant Biotechnology Co., Ltd.), five latex methods (Kit A03, Beijing Jinwofu Bioengineering Technology Co., Ltd.; Kit A04, Hangzhou ACON Biotech Co., Ltd.; Kit A05, Hangzhou AllTest Biotech Co. Ltd.; Kit A06, Tangshan Innovita Biological Technology Co., Ltd.; and Kit A07, Zhuhai Livzon Diagnostics, Inc.), and 12 colloidal gold methods (Kit A08, Guangzhou Wondfo Biotech Co. Ltd.; Kit A09, Nanjing Vazyme Biotech Co., Ltd.; Kit A10, Beijing Hotgen Biotech Co., Ltd.; Kit A11, Beijing Lepu Medical Technology Co., Ltd.; Kit A12, Beijing Wantai Biological Pharmacy Enterprise Co., Ltd.; Kit A13, Zhejiang Orient Gene Biotech Co., Ltd.; Kit A14, Wuhan EasyDiagnosis Biomedicine Co., Ltd.; Kit A15, Zybio, Inc.; Kit A16, Xiamen AmonMed Biotechnology Co., Ltd.; Kit A17, Shandong Kanghua Biotech Co., Ltd.; Kit A18, Shenzhen YHLO Biotech Co. Ltd.; and Kit A19, Beijing Applied Biological Technologies Co. Ltd.) (Table 1). 50 $\mu$L of the WT, Delta, and Omicron supernatants at $1 \times 10^8$, $5 \times 10^7$, $2.5 \times 10^7$, $1.25 \times 10^7$, $6.25 \times 10^6$, $3.12 \times 10^6$, $1.56 \times 10^6$, $7.81 \times 10^5$, and $3.91 \times 10^5$ copies/mL were added to the sample extraction solution provided with each kit in triplicate, respectively. Also, 50 $\mu$L of the WT, Delta AY.2, Delta AY.1/AY.3, Omicron BA.1, and Omicron BA.2 recombinant N proteins at 1,000, 500, 250, 125, 62.5, 31.25, 15.62, and 7.81 ng/mL were subjected to the same detection processes. The recommended volume of sample extraction solution varied from 280 $\mu$L to 600 $\mu$L for the different kits, resulting in different predilution ratios, which ranged from 1:6.6 to 1:13. The results were independently assessed by two laboratory technicians. In cases of inconsistent events, a third technician was consulted to draw a conclusion.

**Statistical analysis.** The concentration equivalent of the 50 $\mu$L sample was the lowest concentration that achieved a 100% detection rate of the 50 $\mu$L original sample before being added to the sample preservation solution. The lowest concentration with a 100% detection rate was that of the diluted sample. The predilution ratios varied according to the volumes of sample preservation solution (3 mL for the single sampling tests and 12 mL for the 20-in-1 pooling tests) or extraction solution (point-of-care test and rapid antigen tests), to which 50 $\mu$L samples of known concentrations were added (Table 2; Supplemental Material File 2).

The variance between the different tests was compared using Pearson's chi-square test in the SPSS Statistics for Windows (version 19.0; IBM Corp., Armonk, NY, USA) software package. A $P$ value of less than 0.05 was regarded as indicative of a statistically significant result.

## SUPPLEMENTAL MATERIAL

Supplemental material is available online only.
**SUPPLEMENTAL FILE 1**, PDF file, 0.1 MB.
**SUPPLEMENTAL FILE 2**, XLSX file, 0.1 MB.

## ACKNOWLEDGMENTS

The work was supported by the National Key Research and Development Program of China, grant 2021YFC0863300 (J. L.).

We thank Sinovac Biotech Co., Ltd. (China) and Biological, Inc. (China) for providing the inactivated SARS-CoV-2 cell culture supernatants and the recombinant SARS-CoV-2 N proteins, respectively. We thank all of the reagent manufacturers that provided the automatic nucleic acid extraction system, rRT-PCR kits, nucleic acid POCT kits, and rapid antigen test kits for the detection of SARS-CoV-2 in this study.

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
