## [Reviewer comments · Microbiology Spectrum]

Microbiology Spectrum

Evaluation of four strategies for SARS-CoV-2 detection: characteristics and prospects

Rui Zhang, Yuqing Chen, Yu Ma, Yanxi Han, Zhenli Diao, Lu Chang, and Jinming Li

Corresponding Author(s): Rui Zhang, Graduate School, Peking Union Medical College, Chinese Academy of Medical Sciences, Beijing, People's Republic of China; National Center for Clinical Laboratories, Beijing Hospital

Review Timeline:

Submission Date:	June 7, 2022
Editorial Decision:	August 14, 2022
Revision Received:	September 2, 2022
Accepted:	September 27, 2022

Editor: Sen Pei

Reviewer(s): Disclosure of reviewer identity is with reference to reviewer comments included in decision letter(s). The following individuals involved in review of your submission have agreed to reveal their identity: Luciana Jesus Costa (Reviewer #2)

Transaction Report:

DOI: <https://doi.org/10.1128/spectrum.02143-22>

August 14, 2022

Dr. Rui Zhang

Graduate School, Peking Union Medical College, Chinese Academy of Medical Sciences, Beijing, People's Republic of China;
National Center for Clinical Laboratories, Beijing Hospital
Beijing
China

Re: Spectrum02143-22 (Evaluation of four strategies for SARS-CoV-2 detection: characteristics and prospects)

Dear Dr. Rui Zhang:

Thank you for submitting your manuscript to Microbiology Spectrum. Your manuscript has been reviewed by two experts. Based on the review comments, I would like to invite a revision to address the questions raised by the reviewers. Particularly, please elaborate on the methods and presentation of the results.

Link Not Available

Sincerely,

Sen Pei

Journals Department
Reviewer comments:

Reviewer #1 (Comments for the Author):

General comments

The authors performed the study to compare the performance of different nucleic acid amplification tests (NAATs) and rapid antigen tests. A total of six NAATs were evaluated, five of them were real-time RT-PCR assays and the remaining one was POCT. Two sample pooling strategies were assessed for the real-time RT-PCR assays, (1) no pooling and (2) 20-in-1 pooling. A total of 19 rapid antigen kits were evaluated, all of them were based on lateral flow principles. The research gaps were clear, the study was well designed and performed. This parallel comparison could contribute to the diagnostic fields of SARS-CoV-2.

However, the methods used and results presented should be elaborated. I have several comments/queries to the current version of the manuscript. The authors should respond them one by one before the manuscript is accepted for publication.

Major comments

- line 142: need to describe the purposes of the two systems used, (1) RT-qPCR, (2) ddPCR. (1) for reference NAAT test? (2) for quantifying the viral load in copies/mL? If (1) was used as the reference, the results should also be presented.
- line 149: need to mention explicitly here, 'The recombinant N proteins were measured'.
- line 158: to be clear, insert the term 'serial two-fold dilution'
- line 161: need to elaborate the two dilution factors, 1:61 and 1:241 as 50uL in 3050uL and 50uL in 12050uL respectively
- lines 167-169: add numbers 1-5 before describing the name of each assay
- Table 1: need to add the two sub-headings (1) real-time RT-PCR and (2) POCT
- lines 172-178: need to briefly describe the principles and hands-on procedures, not everyone use this POCT. as you quote ID NOW as an example (ref 32) in line 301, you may use this technique to introduce your POCT.
- lines 179-191: add numbers 1-19 before describing the name of each assay, need to briefly describe the principles of the rapid antigen tests. (1) all were based on lateral flow principles, (2) antigen-antibody immune-complexes were based on fluorescence, latex, colloidal gold, (3) need instrument to read results or interpreted by naked eye
- Table 1: need to assign number 1-19 for all rapid antigen tests, the number used should be concordant to those mentioned in the text, lines 179-191 and Table 2.
- line 199: 'two laboratories', typo error? should be laboratory technicians?, you mentioned that your study was a single-centre evaluation (line 133)
- lines 201-203: the author have to use examples to illustrate how to calculate the values of (1) lowest concentration, (2) concentration equivalent to 50uL, I suggest one example (e.g. Daan) from NAAT and one example for rapid antigen test (e.g. test 1), if there is limited of words required by the journal, these illustrations maybe put into supplementary section.
- the authors have to provide the raw data in supplementary section:
 - (1) Rt-pcr no pooling, list out the test results for the 9 dilution points
 - (2) Rt-pcr with pooling, list out the test results for the 9 dilution points
 - (3) POCT, list out the test results for the 9 dilution points
 - (4) rapid antigen tests, supernatants, list out the test results for the 9 dilution points
 - (5) rapid antigen tests, recombinant N proteins, list out the test results for the 8 dilution points
- Table 1, need to explain the ways in measuring the volume of extraction solution, mentioned by the kit insert? As far as I know, the extraction solution used by the rapid antigen kits were not mentioned from the kit insert explicitly. obtained from suppliers? measured by pipettes?
- figure 2: I have no idea how can you come up this figure, you either elaborate the methods in the main text or delete it
- discussion section: need to discuss the sensitivity difference between RT-PCR, POCT and rapid antigen tests for the results obtained in this evaluation

Minor comments

- line 68: need to mention the time for the first detection of the two variants in your country.
- the authors forgot to cite table 4 in the main text

Reviewer #2 (Comments for the Author):

The study by Chen and co-workers is well conducted and relevant as a reference for SARS-CoV-2 diagnosis by a more efficient strategy, especially since it demonstrates the efficacy of several RT-qPCR and antigen tests for Omicron detection. The introduction section is lengthy, but results description is very limited, the variations in detection from different tests is not explored (this is especially relevant for the antigen tests in which a great variation can be observed in tables) and the figure 2 besides being hard to follow, the way it is presented makes it difficult to have a comprehensive appreciation of the results. Authors should say how the limit of detection for each test was calculated.

Minor issues:

- 1) Table 4 is not mentioned withing the text
- 2) The sentence: "Mutations can occur continuously during virus transmission..."(lane 63) should be rephrased to: "naturally occurring mutations can be selected continuously during virus transmission..."

Staff Comments:

Preparing Revision Guidelines

To submit your modified manuscript, log onto the eJP submission site at <https://spectrum.msubmit.net/cgi-bin/main.plex>. Go to Author Tasks and click the appropriate manuscript title to begin the revision process. The information that you entered when you

first submitted the paper will be displayed. Please update the information as necessary. Here are a few examples of required updates that authors must address:

Please return the manuscript within 60 days; if you cannot complete the modification within this time period, please contact me. If you do not wish to modify the manuscript and prefer to submit it to another journal, please notify me of your decision immediately so that the manuscript may be formally withdrawn from consideration by Microbiology Spectrum.

The study by Chen and co-workers is well conducted and relevant as a reference for SARS-CoV-2 diagnosis by a more efficient strategy, especially since it demonstrates the efficacy of several RT-qPCR and antigen tests for Omicron detection. The introduction section is lengthy, but results description is very limited, the variations in detection from different tests is not explored (this is especially relevant for the antigen tests in which a great variation can be observed in tables) and the figure 2 besides being hard to follow, the way it is presented makes it difficult to have a comprehensive appreciation of the results. Authors should say how the limit of detection for each test was calculated.

Minor issues:

- 1) Table 4 is not mentioned withing the text
- 2) The sentence: "Mutations can occur continuously during virus transmission..."(lane 63) should be rephrased to: "naturally occurring mutations can be selected continuously during virus transmission..."

Response to Reviewers

Dear Dr. Pei,

Thank you very much for your email dated August 14, 2022. We have revised the manuscript according to the comments of the reviewers, and the amendments are highlighted in RED in the “Marked Up Manuscript - For Review Only”. We also responded to the comments as listed below. The comments are all valuable and very helpful for improving our paper. We would like to thank the reviewers for the constructive and positive comments.

With best wishes,

Yours sincerely,

Rui Zhang

Replies to the reviewers' comments:

Reviewer #1:

Comment 1: line 142: need to describe the purposes of the two systems used, (1) RT-qPCR, (2) ddPCR. (1) for reference NAAT test? (2) for quantifying the viral load in copies/mL? If (1) was used as the reference, the results should also be presented.

Answer 1: Thank you for your valuable suggestion.

The purpose of RT-qPCR was to perform preliminary quantification based on Ct values and to derive the appropriate sample dilution for ddPCR quantification, while the purpose of ddPCR was to perform absolute quantification based on droplets. The samples were first subjected to a ten-fold serial dilution in PBS, then the viral RNAs were extracted with the QIAamp Viral RNA Mini Kit and reverse transcribed into cDNAs with the PrimeScript RT reagent kit. Subsequently, RT-qPCR was performed to obtain the Ct values of diluted samples and approximate amounts of total RNAs or cDNAs. Then the samples with proper dilutions were subjected to ddPCR analysis.

Considering the absolute quantification of samples was primarily performed using the ddPCR method, we have removed the RT-qPCR method from the main text to eliminate confusion about virus quantification. Therefore, we revised the sentence to “For quantification, the viral RNAs **with proper dilutions** were extracted with the QIAamp Viral RNA Mini Kit (Qiagen, Hilden, Germany) and reverse transcribed into cDNAs with the PrimeScript RT reagent kit (Perfect Real Time; TaKaRa, Japan). Subsequently, **the samples were subjected to droplet digital PCR (ddPCR)** on the Bio-Rad QX-200 System (USA) using the N gene assay recommended by China’s Centers for Disease Control (CDC).” in the “Marked Up Manuscript - For Review Only” (Introduction, Lines 111-117).

Comment 2: line 149: need to mention explicitly here, 'The recombinant N proteins were measured!'

Answer 2: Thank you for your valuable suggestion.

We have revised the sentence to “**The concentrations of recombinant N proteins were measured** using a Qubit 3.0 Fluorometer with the Qubit Protein Assay Kit (Thermo Fisher Scientific, USA).” in the “Marked Up Manuscript - For Review Only” (Introduction, Lines 121-123).

Comment 3: line 158: to be clear, insert the term 'serial two-fold dilution'.

Answer 3: Thank you for your valuable suggestion.

We have revised the sentence to “After full oscillation and blending, **serial two-fold dilutions of the WT, Delta and Omicron cell culture supernatants were performed (5×10^5 , 2.5×10^5 , 1.25×10^5 , 6.25×10^4 , 3.12×10^4 , 1.56×10^4 , 7.81×10^3 , 3.91×10^3 and 1.95×10^3 copies/mL)** and 50 μ L of them were added to each sample, respectively.” in the “Marked Up Manuscript - For Review Only” (Introduction, Lines 128-132).

Comment 4: line 161: need to elaborate the two dilution factors, 1:61 and 1:241 as 50 μ L in 3050 μ L and 50 μ L in 12050 μ L respectively.

Answer 4: Thank you for your valuable suggestion.

We have revised the sentence to “Remarkably, this step with positive supernatants diluted in sample preservation solution introduces a pre-dilution effect (**1:61 as 50 μ L in 3050 μ L for the single sampling test and 1:241 as 50 μ L in 12050 μ L** for the 20-in-1 pooling test), leading to a loss of sensitivity as claimed by the reagent manufacturers.” in the “Marked Up Manuscript - For Review Only” (Introduction, Lines 132-135).

Comment 5: lines 167-169: add numbers 1-5 before describing the name of each assay.

Answer 5: Thank you for your valuable suggestion.

For ease of communication throughout the article, each real-time RT-PCR kit was assigned a code from Kit N01 to Kit N05, POCT was encoded as Kit P01, and each rapid antigen test was assigned a code from Kit A01 to Kit A19. “N”, “P” and “A” are the initials of nucleic acid, POCT and antigen, respectively. We have also replaced the kit names in tables and in the Results and Discussion sections with their respective codes.

Therefore, we have revised the sentence to “**Kit N01:** Daan Gene Co., Ltd. of Sun Yat-sen University, **Kit N02:** Shanghai BioGerm Medical Technology Co., Ltd., **Kit N03:** Sansure Bio-tech Co., Ltd., **Kit N04:** Shanghai Liferiver BioTech Co., Ltd., and **Kit N05:** Wuhan EasyDiagnosis Biomedicine Co., Ltd.” in the “Marked Up Manuscript - For Review Only” (Introduction, Lines 140-143).

Comment 6: Table 1: need to add the two sub-headings (1) real-time RT-PCR and (2) POCT.

Answer 6: Thank you for your valuable suggestion.

We have added two sub-headings “**Real-time RT-PCR tests**” and “**Point-of-care test**” in the “Marked Up Manuscript - For Review Only” (Table 1). Real-time RT-PCR tests included Daan, BioGerm, Sansure, Liferiver and EasyDiagnosis. The point-of-care test included Ustar only.

Comment 7: lines 172-178: need to briefly describe the principles and hands-on procedures, not everyone use this POCT. as you quote ID NOW as an example (ref 32) in line 301, you may use this technique to introduce your POCT.

Answer 7: Thank you for your valuable suggestion.

We have added the principles and procedures of POCT with ID NOW as the reference in the “Marked Up Manuscript - For Review Only” (Introduction, Lines 145-154):
Nucleic acid POCTs are portable, easy-to-operate, and isothermal amplification-based devices, mainly characterized by relative sensitivity, simple operation, short turnaround time, and independence of laboratory settings (18). The EasyNAT COVID-19 RNA Test from Ustar Biotechnologies Ltd. carries out cross priming amplification (CPA) reactions through specific primers, probes, and DNA polymerase with high strand displacement activity to qualitatively detect the ORF1ab gene of SARS-CoV-2 with a claimed LOD of 3×10^3 copies/mL (19). Self-collected nasopharyngeal swabs are immersed in the lysis buffer, one drop of which is added to the module preloaded with nucleic acid amplification reagents. After 55 minutes of reaction, the results can be available in the corresponding lateral flow strips.

Reference:

18. Kortüm S, Krause M, Ott H-J, Kortüm L, Schlautd H-P. Molecular point-of-care testing for SARS-CoV-2 using the ID NOW™ System in Emergency Department: Prospective Evaluation and Implementation in the Care Process <https://doi.org/10.1101/2021.09.09.21263266>.
19. Xu G, Hu L, Zhong H, Wang H, Yusa S, Weiss TC, Romaniuk PJ, Pickerill S, You Q. 2012. Cross priming amplification: mechanism and optimization for isothermal DNA amplification. *Sci Rep* 2:1–7.

Comment 8: lines 179-191: add numbers 1-19 before describing the name of each assay, need to briefly describe the principles of the rapid antigen tests. (1) all were based on lateral flow principles, (2) antigen-antibody immune-complexes were based on fluorescence, latex, colloidal gold, (3) need instrument to read results or interpreted by naked eye.

Answer 8: Thank you for your valuable suggestion.

We have added numbers 1-19 before describing the name of each assay in the “Marked Up Manuscript - For Review Only” (Introduction, Lines 164-177): We included nineteen rapid antigen tests approved by China NMPA in our study, including two fluorescence immunochromatography methods (**Kit A01:** BGI Biotech Co., Ltd. and **Kit A02:** Beijing Savant Biotechnology Co., Ltd.), five latex methods (**Kit A03:** Beijing Jinwofu Bioengineering Technology Co., Ltd., **Kit A04:** Hangzhou ACON Biotech Co., Ltd., **Kit A05:** Hangzhou AllTest Biotech Co. Ltd., **Kit A06:** Tangshan Innovita Biological Technology Co., Ltd. and **Kit A07:** Zhuhai Livzon Diagnostics Inc.) and twelve colloidal gold methods (**Kit A08:** Guangzhou Wondfo Biotech Co. Ltd., **Kit A09:** Nanjing Vazyme Biotech Co., Ltd., **Kit A10:** Beijing Hotgen Biotech Co., Ltd., **Kit A11:** Beijing Lepu Medical Technology Co., Ltd., **Kit A12:** Beijing Wantai Biological Pharmacy Enterprise Co., Ltd., **Kit A13:** Zhejiang Orient Gene Biotech Co., Ltd., **Kit A14:** Wuhan EasyDiagnosis Biomedicine Co., Ltd., **Kit A15:** Zybio, Inc., **Kit A16:** Xiamen AmonMed Biotechnology Co., Ltd., **Kit A17:** Shandong Kanghua Biotech Co., Ltd., **Kit A18:** Shenzhen YHLO Biotech Co. Ltd., and **Kit A19:** Beijing Applied Biological Technologies Co. Ltd.).

And we have added the principles of rapid antigen tests in the “Marked Up Manuscript - For Review Only” (Introduction, Lines 159-164): **The main principle behind rapid antigen tests is the use of lateral flow immunoassays designed with the fluorescence microsphere, latex microsphere, or colloidal gold labeled SARS-CoV-2 protein antibody to form an antibody-antigen (Ab-Ag) complex. The test results can be interpreted by simple instruments for the fluorescence method or naked eyes for the latex and colloidal gold methods within 15 minutes.**

Comment 9: Table 1: need to assign numbers 1-19 for all rapid antigen tests, the number used should be concordant to those mentioned in the text, lines 179-191 and Table 2.

Answer 9: Thank you for your valuable suggestion.

We have added a column named “Code” to Table 1 in order to code each detection kit in the “Marked Up Manuscript - For Review Only”. The codes were concordant with those mentioned in the main text and were used for the following Results and Discussion sections. Therefore, we have added the footnote in Table 1: **For ease of communication throughout the article, each real-time RT-PCR kit was assigned a code from Kit N01 to Kit N05, POCT was encoded as Kit P01, and each rapid antigen test was assigned a code from Kit A01 to Kit A19.** And we have replaced the kit names in the other tables and in the Results and Discussion sections with their respective codes.

Comment 10: line 199: 'two laboratories', typo error? should be laboratory technicians?, you mentioned that your study was a single-centre evaluation (line 133).

Answer 10: Thank you for your valuable suggestion.

We intended to mean two laboratory technicians and we have corrected the mistake in the “Marked Up Manuscript - For Review Only” (Introduction, Lines 185-186): The results were independently assessed by **two laboratory technicians**.

Comment 11: lines 201-203: the authors have to use examples to illustrate how to calculate the values of (1) lowest concentration, (2) concentration equivalent to 50uL, I suggest one example (e.g. Daan) from NAAT and one example for rapid antigen test (e.g. test 1), if there is limited of words required by the journal, these illustrations maybe put into supplementary section.

Answer 11: Thank you for your valuable suggestion.

For the single sampling tests and 20-in-1 pooling tests, 50 µL cell culture supernatants were added to 3 mL and 12 mL sample preservation solution, resulting in 61-fold and 241-fold dilutions of the initial sample concentration, respectively. The “concentration equivalent to the 50 µL sample” in Table 3 was the lowest concentration to achieve 100% detection rates of the 50 µL original sample before being added to the sample preservation solution. The “lowest concentration at 100% detection rates” was the lowest concentration to achieve 100% detection rates of the diluted sample after being added to the sample preservation solution, and was 1/61 (single sampling tests) or 1/241 (20-in-1 pooling tests) of the 50 µL original concentration.

For the point-of-care test, 50 µL cell culture supernatants were added to 500 µL sample extraction solution, resulting in an 11-fold dilution of the initial sample concentration. Similar to the rRT-PCR kits, the “lowest concentration at 100% detection rates” was 1/11 of the “concentration equivalent to the 50 µL sample” in Table 3.

For the rapid antigen tests, 50 µL cell culture supernatants or recombinant N proteins were added to different volumes of sample extraction solution for different antigen tests. Taking BGI as an example, 50 µL cell culture supernatants or recombinant N proteins were added to 400 µL sample extraction solution resulting in a 9-fold dilution of the initial sample concentration. Therefore, the “lowest concentration at 100% detection rates” was 1/9 of the “concentration equivalent to the 50 µL sample” in Table 3 and Table 4.

In order to describe the pre-dilution ratio and the calculation of the LOD for each kit more clearly, we have added “**Table 2 Calculation of pre-dilution ratio of different detection methods.**”

Table 2 Calculation of pre-dilution ratio of different detection methods.

Detection kits	Volume of Sample original	of Sample solution/Sample	preservation solution	Final volume	Pre-dilution ratio
----------------	---------------------------	---------------------------	-----------------------	--------------	--------------------

	samples	solution ^a		
Real-time RT-PCR tests				
Kit N01	50 µL			
Kit N02	50 µL		Single sampling	
Kit N03	50 µL	Single sampling test: 3 mL;	test: 3.05 mL;	Single sampling test: 61;
Kit N04	50 µL	20-in-1 pooling test: 12 mL	20-in-1 pooling	20-in-1 pooling test: 241
Kit N05	50 µL		test: 12.05 mL	
Point-of-care test				
Kit P01	50 µL	500 µL	550 µL	11
Rapid antigen tests				
Kit A01	50 µL	400 µL	450 µL	9
Kit A02	50 µL	500 µL	550 µL	11
Kit A03	50 µL	500 µL	550 µL	11
Kit A04	50 µL	340 µL	390 µL	7.8
Kit A05	50 µL	300 µL	350 µL	7
Kit A06	50 µL	300 µL	350 µL	7
Kit A07	50 µL	600 µL	650 µL	13
Kit A08	50 µL	400 µL	450 µL	9
Kit A09	50 µL	500 µL	550 µL	11
Kit A10	50 µL	300 µL	350 µL	7
Kit A11	50 µL	400 µL	450 µL	9
Kit A12	50 µL	500 µL	550 µL	11
Kit A13	50 µL	300 µL	350 µL	7
Kit A14	50 µL	280 µL	330 µL	6.6
Kit A15	50 µL	500 µL	550 µL	11
Kit A16	50 µL	350µL	400µL	8
Kit A17	50 µL	500µL	550µL	11
Kit A18	50 µL	400µL	450µL	9

^aThe volumes of extraction solution were obtained from the instructions or reagent manufacturers and verified by manual pipettes.

Besides, two examples from Daan and BGI were attached in the Supplementary Material File 2 to describe the calculation method in detail:

For the single sampling tests and 20-in-1 pooling tests, 50 μ L cell culture supernatants were added to 3 mL and 12 mL sample preservation solution. The concentration equivalent to the 50 μ L sample was the lowest concentration to achieve 100% detection rates of the 50 μ L original sample before being added to the sample preservation solution. The lowest concentration at 100% detection rates of the diluted sample was 1/61 (single sampling tests, 50 μ L in 3050 μ L) or 1/241 (20-in-1 pooling tests, 50 μ L in 12050 μ L) of the 50 μ L original concentration. Assuming that the concentration equivalent to the 50 μ L sample for the single sampling tests using the Daan kit was 6.25×10^4 copies/mL, then the lowest concentration at 100% detection rates was 6.25×10^4 divided by 61, namely 1.02×10^3 copies/mL.

For the point-of-care test and rapid antigen tests, the multiples varied according to the amount of extraction solution of different kits. Taking the antigen test BGI as an example, the volume of extraction solution was 400 μ L and the lowest concentration at 100% detection rates was 1/9 (50 μ L in 450 μ L) of the concentration equivalent to the 50 μ L sample. Assuming that the concentration equivalent to the 50 μ L sample for the BGI test was 3.12×10^6 copies/mL, then the lowest concentration at 100% detection rates was 3.12×10^6 divided by 9, namely 3.47×10^5 copies/mL.

Therefore, we have added the sentences in the “Marked Up Manuscript - For Review Only” (Introduction, Lines 188-195): The concentration equivalent to the 50 μ L sample was the lowest concentration to achieve 100% detection rates of the 50 μ L original sample before being added to the sample preservation solution. The lowest concentration at 100% detection rates was that of the diluted sample. The pre-dilution ratios varied according to the volume of sample preservation solution (3 mL for the

single sampling tests and 12 mL for the 20-in-1 pooling tests) or extraction solution (point-of-care test and rapid antigen tests) to which 50 μ L samples of known concentrations were added (Table 2, Supplementary Material File 2).

Comment 12: The authors have to provide the raw data in supplementary section:

- (1) Rt-pcr no pooling, list out the test results for the 9 dilution points
- (2) Rt-pcr with pooling, list out the test results for the 9 dilution points
- (3) POCT, list out the test results for the 9 dilution points
- (4) rapid antigen tests, supernatants, list out the test results for the 9 dilution points
- (5) rapid antigen tests, recombinant N proteins, list out the test results for the 8 dilution points

Answer 12: Thank you for your valuable suggestion.

We have added Supplementary Material File 3 attached with the raw data of test results. Sheet 1 included the original Ct values and positive rates of 9 dilution concentrations of WT, Delta and Omicron strains for the single sampling tests.

Sheet 2 included the original Ct values and positive rates of 9 dilution concentrations of WT, Delta and Omicron strains for the 20-in-1 pooling tests.

Sheet 3 included the positive rates of 9 dilution concentrations of WT, Delta and Omicron strains for the point-of-care test.

Sheet 4 included the positive rates of 9 dilution concentrations of WT, Delta and Omicron inactivated cell culture supernatants as well as 8 dilution concentrations of WT, Delta AY.2, Delta AY.1/AY.3, Omicron BA.1 and Omicron BA.2 recombinant N proteins for the rapid antigen tests.

Comment 13: Table 1, need to explain the ways in measuring the volume of extraction solution, mentioned by the kit insert? As far as I know, the extraction

solution used by the rapid antigen kits were not mentioned from the kit insert explicitly. obtained from suppliers? measured by pipettes?

Answer 13: Thank you for your valuable suggestion.

Fifteen manufacturers specified the volume of extraction solution in their instructions. For the remaining four antigen kits (Jinwofu, ACON, Orient Gene and Zybion) without mentioning the volume of extraction solution, we would consult the reagent manufacturers and verify by manual pipettes.

Because we added Table 2 to the article, and the volumes of extraction solution of different rapid antigen tests were moved to Table 2. Therefore, we added the footnote in Table 2: **The volumes of extraction solution were obtained from the instructions or reagent manufacturers and verified by manual pipettes.**

Comment 14: figure 2: I have no idea how can you come up this figure, you either elaborate the methods in the main text or delete it.

Answer 14: Thank you for your valuable suggestion.

To reduce unnecessary confusion, we decided to use tables mainly to present the results and delete Figure 2. Table 3 and Table 4 showed the lowest concentrations of samples to achieve 100% detection rates. Besides, to provide more detailed information in the study, we have added Supplementary Material File 3 attached with the raw data of test results.

Comment 15: discussion section: need to discuss the sensitivity difference between RT-PCR, POCT and rapid antigen tests for the results obtained in this evaluation

Answer 15: Thank you for your valuable suggestion.

We supplemented the discussion for the sensitivity difference between RT-PCR, POCT and rapid antigen tests. Correspondingly, the test results were analyzed in Results according to the comment of another reviewer. The revisions are as follows:

(1) Real-time RT-PCR

Results: At first, we compared the sensitivity difference between the single sampling tests and 20-in-1 pooling tests. As shown in the Results section, for single sampling tests and five rRT-PCR detection kits, the lowest inactivated virus concentrations to achieve 100% detection ranged between 1.28×10^2 - 1.02×10^3 , 1.28×10^2 - 4.10×10^3 , and 1.28×10^2 - 2.05×10^3 copies/mL for the WT, Delta and Omicron strains, respectively (Results, Lines 202-206). For 20-in-1 pooling tests and five rRT-PCR detection kits, the lowest concentrations ranged between 1.30×10^2 - 1.04×10^3 , 5.19×10^2 - 2.07×10^3 , and 2.59×10^2 - 1.04×10^3 copies/mL for the WT, Delta and Omicron strains, respectively (Results, Lines 207-210). Overall, when detecting the diluted samples higher than the claimed LODs, the 20-in-1 pooling tests could substantially achieve the same detection performance as the single sampling tests. When it came to Delta (22/27 vs 15/27, P = 0.040) and Omicron (22/27 vs 14/27, P = 0.021) variants lower than the claimed LODs, the impaired analytical sensitivity was found in the 20-in-1 pooling strategy using Kit N03 (Results, Lines 215-219).

Discussion: Therefore, we discussed the results in the “Marked Up Manuscript - For Review Only”: When using five rRT-PCR kits to detect the same strain, the range of lowest inactivated virus concentrations to achieve 100% detection by 20-in-1 pooling tests was substantially the same as single sampling tests. Overall, when detecting the diluted samples higher than the claimed LODs, 20-in-1 pooling tests showed comparable analytical sensitivity to the single sampling tests (Discussion, Lines 298-302). Therefore, the 20-in-1 pooling tests enable the mass nucleic acid tests with only 5% of the original testing workload, greatly improving daily testing efficiency, which is the first consideration in the screening of asymptomatic infected cases in low-risk regions (Discussion, Lines 304-307).

(2) Point-of care test

Results: Unexpectedly, the Kit P01 showed comparable analytical sensitivity to rRT-PCR kits with the lowest concentration of 1.42×10^3 copies/mL for all three strains in the detection of SARS-CoV-2 in the study (Results, Lines 212-214).

Discussion: Therefore, we concluded in the “Marked Up Manuscript - For Review Only”: With short turnaround time, portable procedures, and analytical sensitivity comparable to the gold standard rRT-PCR, nucleic acid POCT is a promising diagnostic agent for aiding testing expansion and can be used as confirmatory testing where laboratory-based nucleic acid amplification test is not available (Discussion, Lines 323-327).

(3) Rapid antigen tests

Results: For rapid antigen tests, on testing of inactivated cell culture supernatants, the lowest virus concentrations to achieve a 100% detection rate ranged between 2.84×10^5 - 7.14×10^6 , 8.68×10^4 - 7.14×10^6 , and 1.12×10^5 - 3.57×10^6 copies/mL for the WT, Delta and Omicron strains, respectively, corresponding to 3.12×10^6 - 5.00×10^7 , 7.81×10^5 - 5.00×10^7 , and 7.81×10^5 - 2.50×10^7 copies/mL of the 50 μ L original sample (Results, Lines 224-228). And the original concentrations of the 50 μ L samples to achieve 100% detection for the single sampling tests were 7.81×10^3 - 6.25×10^4 , 7.81×10^3 - 2.50×10^5 , and 7.81×10^3 - 1.25×10^5 copies/mL (Results, Lines 206-207).

Different from the rRT-PCR kits, great variations in analytical sensitivity were observed among rapid antigen kits. The lowest concentrations that achieved a 100% rate of detection success ranged from 10^4 to 10^6 copies/mL for inactivated cell culture supernatants and from 1 to 150 ng/mL for recombinant N proteins (Results, Lines 236-239).

Discussion: Therefore, we discussed the results in the “Marked Up Manuscript - For Review Only”: Our study revealed that the detection performance of different rapid antigen kits varied greatly, and the lowest concentrations that achieved a 100% rate of detection success ranged from 10^4 to 10^6 copies/mL for inactivated cell culture

supernatants and from 1 to 150 ng/mL for recombinant N proteins, which was essentially consistent with previously published articles. The antibody labeling method of rapid antigen tests would affect their analytical performance, and the fluorescence microsphere was observed to enhance the sensitivity of analytical signal by 10 to 100 folds compared to latex microsphere and colloidal gold (40). Besides, the lowest concentrations of the 50 µL original sample at 100% detection of rapid antigen tests (10^5 - 10^7 copies/mL) were about 100 times as high as those of the single sampling test (10^3 - 10^5 copies/mL) (Discussion, Lines 336-346). On the whole, although not as sensitive and specific as molecular tests for the diagnosis of SARS-CoV-2 infection, the rapid antigen tests should be considered as a rapid diagnostic tool with minimal training and cost-effective and time-saving detection processes for instantaneous assessment of infectivity rather than exclusion of infection, mass screening in high-risk areas, and detecting symptomatic cases meeting the COVID-19 case definition (Discussion, Lines 352-357).

Reference:

40. Hampl J, Hall M, Mufti NA, Yung-mae MY, MacQueen DB, Wright WH, Cooper DE. 2001. Upconverting phosphor reporters in immunochromatographic assays. Anal Biochem 288:176–187.

(4) Test results among variants:

Besides, we discussed the potential effect of virus mutations on test results.

Results: For nucleic acid amplification tests, contrasted with the WT strain, the impaired analytical sensitivity was found in the Delta variant using Kit N04 both for the single sampling tests (19/27 vs 11/27, $P = 0.028$) and the 20-in-1 pooling tests (16/27 vs 7/27, $P = 0.013$). And the Omicron variant had no significant effect on the five rRT-PCR kits and POCT kit (Results, Lines 220-223). For each rapid antigen test, no significant difference in analytical sensitivity was found among three inactivated cell culture supernatants or five recombinant N proteins (Results, Lines 243-245).

Discussion: Therefore, we discussed the results in the “Marked Up Manuscript - For Review Only” (Discussion, Lines 364-379): The continuous evolution of SARS-CoV-2 also affects the accuracy and reliability of detection results to a certain extent. Once the mutations of the virus are located at the primer/probe-targeted regions, the effectiveness of molecular assays might be influenced to some degree. In this study, contrasted with the WT strain, impaired analytical sensitivity was found in the Delta variant using Kit N04 both for the single sampling tests (19/27 vs 11/27, $P = 0.028$) and the 20-in-1 pooling tests (16/27 vs 7/27, $P = 0.013$), and the Omicron variant had no significant effect on the five rRT-PCR kits and POCT kit, which conformed to the previous study. However, the 19 rapid antigen tests showed a non-significant difference in analytical sensitivity for the three inactivated cell culture supernatants and five recombinant N proteins, which might be attributed to the fact that nearly all of the rapid antigen tests target the SARS-CoV-2 nucleocapsid protein with less possibility to mutate. Hence, to ensure the validity of different detection strategies under the background of SARS-CoV-2 genomic diversity, it is essential to consolidate genomic surveillance and track the potential effect of mutation on detection performance.

In the end, we summarized the sensitivity difference among the four detection strategies obtained in this evaluation and provided helpful insights into their scientific deployment in the last paragraph of the article (Discussion, Lines 382-389): Generally, the analytical sensitivity of nucleic acid amplification tests was superior to that of rapid antigen tests, with single sampling strategy showing the highest. In a specific scenario, the optimal strategy should be adopted in consideration of the testing purpose, resource availability, cost performance and result rapidity on the premise of test accuracy, thereby improving overall detection efficiency, facilitating the discovery of early community transmission, and enabling timely and long-term infection control measures under the condition of limited detection capacity and overburdened laboratory infrastructure.

Comment 16: line 68: need to mention the time for the first detection of the two variants in your country.

Answer 16: Thank you for your valuable suggestion.

We have added the time for the first detection of the two variants in China in the “Marked Up Manuscript - For Review Only” (Introduction, Lines 64-67): Among the Variants of Concern (VOC) and Variants of Interest (VOI) designated by the World Health Organization (WHO) based on the transmissibility, pathogenicity, and threat to public health, **Delta and Omicron variants have been first reported in Guangdong on May 18, 2021 (5) and Tianjin on December 13, 2021 (6), respectively.**

References:

5. Zhang M, Xiao J, Deng A, Zhang Y, Zhuang Y, Hu T, Li J, Tu H, Li B, Zhou Y. 2021. Transmission dynamics of an outbreak of the COVID-19 Delta variant B. 1.617. 2—Guangdong Province, China, May–June 2021. *China CDC Weekly* 3:584.
6. Tan Z, Chen Z, Yu A, Li X, Feng Y, Zhao X, Xu W, Su X. 2022. The First Two Imported Cases of SARS-CoV-2 Omicron Variant — Tianjin Municipality, China, December 13, 2021. *China CDC Weekly*. Chinese Center for Disease Control and Prevention <https://doi.org/10.46234/ccdew2021.266>.

Comment 17: the authors forgot to cite table 4 in the main text

Answer 17: Thank you for your valuable suggestion.

Because we added Table 2 to the article, Table 4 became Table 5 accordingly. We cited Table 5 in the “Marked Up Manuscript - For Review Only” (Discussion, Lines 382): In conclusion, the study evaluated the analytical sensitivity of four

SARS-CoV-2 detection strategies applied in different settings and provided helpful insights into the scientific deployment of these tests (Table 5).

Reviewer #2:

Comment 1: The introduction section is lengthy, but results description is very limited, the variations in detection from different tests is not explored (this is especially relevant for the antigen tests in which a great variation can be observed in tables)

Answer 1: Thank you for your valuable suggestion.

At first, we refined the Introduction section and deleted some sentences about the epidemiology of Delta and Omicron variants and the characteristics of single sampling tests and 20-in-1 pooling tests. Then, we fleshed out the Discussion section and added the following three paragraphs in the appropriate positions of the Discussion section: In China, since May 2021, the Delta variant with an R0 below 7 (original strain 2.5) has ravaged many cities, such as Guangzhou, Nanjing, Yangzhou, Putian, Xiamen, and Ejina Banner. The Omicron variant is estimated to have an R0 up to 10 and a doubling time of every 2–3 days, which makes it reasonable to supersede Delta as the dominant variant by December 2021. With the proportion of asymptomatic infections calculated to be as high as 80-90% as well as the rapid occult transmissibility, the Omicron variant has swept numerous cities with an unprecedented speed, especially in Shanghai, where daily confirmed infected cases have roared up to 20000 for days on end (Discussion, Lines 247-255). At the very beginning of outbreaks, the single sampling test of rRT-PCR was regarded as the most sensitive and specific method in detecting SARS-CoV-2, which was recommended for confirming infected cases and testing specific groups including contacts of confirmed or frequently exposed groups to ensure the timely implementation of public health

measures and patient management procedures such as contact tracing and quarantine (Discussion, Lines 275-280). However, the primary bottleneck of pooled testing is the reduced concentration of viral genetic material below the limit of detection (LOD) for a certain test due to sample dilution, thereby leading to decreased diagnostic sensitivity and false-negative results (Discussion, Lines 295-298).

Next, we further analyzed the test results and supplemented the Results section. Then, the corresponding conclusions were added to the Discussion section.

(1) Real-time RT-PCR

Results: For single sampling tests and five rRT-PCR detection kits, the lowest inactivated virus concentrations to achieve 100% detection ranged between 1.28×10^2 - 1.02×10^3 , 1.28×10^2 - 4.10×10^3 , and 1.28×10^2 - 2.05×10^3 copies/mL for the WT, Delta and Omicron strains, respectively, corresponding to 7.81×10^3 - 6.25×10^4 , 7.81×10^3 - 2.50×10^5 , and 7.81×10^3 - 1.25×10^5 copies/mL of the 50 μ L original sample (Results, Lines 202-207). For 20-in-1 pooling tests and five rRT-PCR detection kits, the lowest concentrations ranged between 1.30×10^2 - 1.04×10^3 , 5.19×10^2 - 2.07×10^3 , and 2.59×10^2 - 1.04×10^3 copies/mL for the WT, Delta and Omicron strains, respectively. Kit N03 reliably detected as low as 10^2 copies/mL for both sampling strategies, showing the most sensitive performance compared with other rRT-PCR kits (Results, Lines 207-212).

Overall, when detecting the diluted samples higher than the claimed LODs, the 20-in-1 pooling tests could substantially achieve the same detection performance as the single sampling tests. When it came to Delta (22/27 vs 15/27, P = 0.040) and Omicron (22/27 vs 14/27, P = 0.021) variants lower than the claimed LODs, the impaired analytical sensitivity was found in the 20-in-1 pooling strategy using Kit N03 (Results, Lines 215-219).

Discussion: Based on the results above, we discussed the results in the Discussion section: When using five rRT-PCR kits to detect the same strain, the range of lowest inactivated virus concentrations to achieve 100% detection by 20-in-1 pooling tests

was substantially the same as single sampling tests. Overall, when detecting the diluted samples higher than the claimed LODs, 20-in-1 pooling tests showed comparable analytical sensitivity to the single sampling tests (Discussion, Lines 298-302). Therefore, the 20-in-1 pooling tests enable the mass nucleic acid tests with only 5% of the original testing workload, greatly improving daily testing efficiency, which is the first consideration in the screening of asymptomatic infected cases in low-risk regions (Discussion, Lines 304-307).

(2) Point-of care test

Results: Unexpectedly, the POCT Kit P01 showed comparable analytical sensitivity to rRT-PCR kits with the lowest concentration of 1.42×10^3 copies/mL for all three strains (Table 2, Supplementary Material File 3) (Results, Lines 212-214).

Discussion: Therefore, we concluded in the Discussion section: With short turnaround time, portable procedures, and analytical sensitivity comparable to the gold standard rRT-PCR, nucleic acid POCT is a promising diagnostic agent for aiding testing expansion and can be used as confirmatory testing where laboratory-based nucleic acid amplification test is not available (Discussion, Lines 323-327).

(3) Rapid antigen tests

Results: On testing of inactivated cell culture supernatants, the lowest virus concentrations to achieve a 100% detection rate ranged between 2.84×10^5 - 7.14×10^6 , 8.68×10^4 - 7.14×10^6 , and 1.12×10^5 - 3.57×10^6 copies/mL for the WT, Delta and Omicron strains, respectively, corresponding to 3.12×10^6 - 5.00×10^7 , 7.81×10^5 - 5.00×10^7 , and 7.81×10^5 - 2.50×10^7 copies/mL of the 50 μ L original sample (Table 3, Supplementary Material File 3). We also tested nineteen rapid antigen tests with five recombinant SARS-CoV-2 N proteins. The lowest concentrations ranged between 3.47-142.86 ng/mL, 1.74-142.86 ng/mL, 3.47-142.86 ng/mL, 3.47-142.86 ng/mL, and 5.68-142.86 ng/mL for the WT, Delta AY.2, Delta AY.1/AY.3, Omicron BA.1 and Omicron BA.2 recombinant N proteins, respectively (Table 4, Supplementary Material File 3). Almost all rapid antigen tests reliably detected

around 2.50×10^7 copies/mL of 50 μ L inactivated supernatants and 500 ng/mL of 50 μ L recombinant N proteins (Results, Lines 224-235).

Different from the rRT-PCR kits, great variations in analytical sensitivity were observed among rapid antigen kits. The lowest concentrations that achieved a 100% rate of detection success ranged from 10^4 to 10^6 copies/mL for inactivated cell culture supernatants and from 1 to 150 ng/mL for recombinant N proteins. The best analytical performance was achieved by Kit A05 in detecting inactivated viruses, while detecting recombinant N proteins, Kit A01 performed the best. The assay manufactured by Kit A19 was considerably less sensitive than the other assays both in detecting inactivated viruses and recombinant N proteins (Results, Lines 236-243).

Discussion: Based on the results above, we concluded in the Discussion section: **Our study revealed that the detection performance of different rapid antigen kits varied greatly**, and the lowest concentrations that achieved a 100% rate of detection success ranged from 10^4 to 10^6 copies/mL for inactivated cell culture supernatants and from 1 to 150 ng/mL for recombinant N proteins, which was essentially consistent with previously published articles. **The antibody labeling method of rapid antigen tests would affect their analytical performance, and the fluorescence microsphere was observed to enhance the sensitivity of analytical signal by 10 to 100 folds compared to latex microsphere and colloidal gold (40)**. Besides, the lowest concentrations of the 50 μ L original sample at 100% detection of rapid antigen tests (10^5 - 10^7 copies/mL) were about 100 times as high as those of the single sampling test (10^3 - 10^5 copies/mL) (Discussion, Lines 336-346). **On the whole, although not as sensitive and specific as molecular tests for the diagnosis of SARS-CoV-2 infection**, the rapid antigen tests should be considered as a rapid diagnostic tool **with minimal training and cost-effective and time-saving detection processes** for instantaneous assessment of infectivity rather than exclusion of infection, mass screening in high-risk areas, and detecting symptomatic cases meeting the COVID-19 case definition (Discussion, Lines 352-357).

Reference:

40. Hampl J, Hall M, Mufti NA, Yung-mae MY, MacQueen DB, Wright WH, Cooper DE. 2001. Upconverting phosphor reporters in immunochromatographic assays. *Anal Biochem* 288:176–187.

(4) Test results among variants

Results: In terms of test results among variants, contrasted with the WT strain, the Delta variant adversely affected the analytical sensitivity of Kit N04 both for the single sampling tests (19/27 vs 11/27, $P = 0.028$) and the 20-in-1 pooling tests (16/27 vs 7/27, $P = 0.013$). And the Omicron variant had no significant effect on the five rRT-PCR kits and POCT kit (Results, Line 219-223). For each rapid antigen test, no significant difference in analytical sensitivity was found among three inactivated cell culture supernatants or five recombinant N proteins (Results, Lines 243-245).

Discussion: Therefore, we discussed the results in the Discussion section (Discussion, Lines 364-379): The continuous evolution of SARS-CoV-2 also affects the accuracy and reliability of detection results to a certain extent. Once the mutations of the virus are located at the primer/probe-targeted regions, the effectiveness of molecular assays might be influenced to some degree. In this study, contrasted with the WT strain, impaired analytical sensitivity was found in the Delta variant using Kit N04 both for the single sampling tests (19/27 vs 11/27, $P = 0.028$) and the 20-in-1 pooling tests (16/27 vs 7/27, $P = 0.013$), and the Omicron variant had no significant effect on the five rRT-PCR kits and POCT kit, which conformed to the previous study. However, the 19 rapid antigen tests showed a non-significant difference in analytical sensitivity for the three inactivated cell culture supernatants and five recombinant N proteins, which might be attributed to the fact that nearly all of the rapid antigen tests target the SARS-CoV-2 nucleocapsid protein with less possibility to mutate. Hence, to ensure the validity of different detection strategies under the background of SARS-CoV-2 genomic diversity, it is essential to consolidate genomic surveillance and track the potential effect of mutation on detection performance.

In the end, we summarized the sensitivity difference among the four detection strategies obtained in this evaluation and provided helpful insights into their scientific deployment in the last paragraph of the article (Discussion, Lines 382-389): **Generally, the analytical sensitivity of nucleic acid amplification tests was superior to that of rapid antigen tests, with single sampling strategy showing the highest.** In a specific scenario, the optimal strategy should be adopted in consideration of the testing purpose, resource availability, cost performance and result rapidity on the premise of test accuracy, thereby improving overall detection efficiency, facilitating the discovery of early community transmission, and enabling timely and long-term infection control measures under the condition of limited detection capacity and overburdened laboratory infrastructure.

Comment 2: figure 2 besides being hard to follow, the way it is presented makes it difficult to have a comprehensive appreciation of the results.

Answer 2: Thank you for your valuable suggestion.

To reduce unnecessary confusion, we decided to use tables mainly to present the results and delete Figure 2. Table 3 and Table 4 showed the lowest concentrations of samples to achieve 100% detection rates. Besides, to provide more detailed information in the study, we have added Supplementary Material File 3 attached with the raw data of test. Besides, to provide more detailed information in the study, we have added **Supplementary Material File 3** attached with the raw data of test results.

Sheet 1 included the original Ct values and positive rates of 9 dilution concentrations of WT, Delta and Omicron variants for the single sampling tests.

Sheet 2 included the original Ct values and positive rates of 9 dilution concentrations of WT, Delta and Omicron variants for the 20-in-1 pooling tests.

Sheet 3 included the positive rates of 9 dilution concentrations of WT, Delta and Omicron variants for the point-of-care test.

Sheet 4 included the positive rates of 9 dilution concentrations of WT, Delta and Omicron inactivated cell culture supernatants as well as 9 dilution concentrations of WT, Delta AY.2, Delta AY.1/AY.3, Omicron BA.1 and Omicron BA.2 recombinant N proteins for the rapid antigen tests.

Comment 3: Authors should say how the limit of detection for each test was calculated.

Answer 3: Thank you for your valuable suggestion.

For the single sampling tests and 20-in-1 pooling tests, 50 μ L cell culture supernatants were added to 3 mL and 12 mL sample preservation solution, resulting in 61-fold and 241-fold dilutions of the initial sample concentration, respectively. The “concentration equivalent to the 50 μ L sample” in Table 3 was the lowest concentration to achieve 100% detection rates of the 50 μ L original sample before being added to the sample preservation solution. The “lowest concentration at 100% detection rates” was the lowest concentration to achieve 100% detection rates of the diluted sample after being added to the sample preservation solution, and was 1/61 (single sampling tests) or 1/241 (20-in-1 pooling tests) of the 50 μ L original concentration.

For the point-of-care test, 50 μ L cell culture supernatants were added to 500 μ L sample extraction solution, resulting in an 11-fold dilution of the initial sample concentration. Similar to the rRT-PCR kits, the “lowest concentration at 100% detection rates” was 1/11 of the “concentration equivalent to the 50 μ L sample” in Table 3.

For the rapid antigen tests, 50 μ L cell culture supernatants or recombinant N proteins were added to different volumes of sample extraction solution for different antigen tests. Taking BGI as an example, 50 μ L cell culture supernatants or recombinant N proteins were added to 400 μ L sample extraction solution resulting in a 9-fold dilution of the initial sample concentration. Therefore, the “lowest concentration at 100%

detection rates” was 1/9 of the “concentration equivalent to the 50 µL sample” in Table 3 and Table 4.

In order to describe the pre-dilution ratio and the calculation of the LOD for each kit more clearly, we have added “Table 2 Calculation of pre-dilution ratio of different detection methods.”

Table 2 Calculation of pre-dilution ratio of different detection methods.

Detection kits	Volume of original samples	of Sample solution/Sample solution ^a	preservation extraction	Final volume	Pre-dilution ratio
Real-time RT-PCR tests					
Kit N01	50 µL			Single sampling	
Kit N02	50 µL				
Kit N03	50 µL		Single sampling test: 3 mL;	test: 3.05 mL;	Single sampling test: 61;
Kit N04	50 µL		20-in-1 pooling test: 12 mL	20-in-1 pooling test: 12.05 mL	20-in-1 pooling test: 241
Kit N05	50 µL				
Point-of-care test					
Kit P01	50 µL	500 µL		550 µL	11
Rapid antigen tests					
Kit A01	50 µL	400 µL		450 µL	9
Kit A02	50 µL	500 µL		550 µL	11
Kit A03	50 µL	500 µL		550 µL	11
Kit A04	50 µL	340 µL		390 µL	7.8
Kit A05	50 µL	300 µL		350 µL	7
Kit A06	50 µL	300 µL		350 µL	7
Kit A07	50 µL	600 µL		650 µL	13
Kit A08	50 µL	400 µL		450 µL	9
Kit A09	50 µL	500 µL		550 µL	11
Kit A10	50 µL	300 µL		350 µL	7

Kit A11	50 µL	400 µL	450 µL	9
Kit A12	50 µL	500 µL	550 µL	11
Kit A13	50 µL	300 µL	350 µL	7
Kit A14	50 µL	280 µL	330 µL	6.6
Kit A15	50 µL	500 µL	550 µL	11
Kit A16	50 µL	350µL	400µL	8
Kit A17	50 µL	500µL	550µL	11
Kit A18	50 µL	400µL	450µL	9
Kit A19	50 µL	300µL	350µL	7

^aThe volumes of extraction solution were obtained from the instructions or reagent manufacturers and verified by manual pipettes.

Besides, two examples from Daan and BGI were attached in the Supplementary Material File 2 to describe the calculation method in detail:

For the single sampling tests and 20-in-1 pooling tests, 50 µL cell culture supernatants were added to 3 mL and 12 mL sample preservation solution. The concentration equivalent to the 50 µL sample was the lowest concentration to achieve 100% detection rates of the 50 µL original sample before being added to the sample preservation solution. The lowest concentration at 100% detection rates of the diluted sample was 1/61 (single sampling tests, 50 µL in 3050 µL) or 1/241 (20-in-1 pooling tests, 50 µL in 12050 µL) of the 50 µL original concentration. Assuming that the concentration equivalent to the 50 µL sample for the single sampling tests using the Daan kit was 6.25×10^4 copies/mL, then the lowest concentration at 100% detection rates was 6.25×10^4 divided by 61, namely 1.02×10^3 copies/mL.

For the point-of-care test and rapid antigen tests, the multiples varied according to the amount of extraction solution of different kits. Taking the antigen test BGI as an example, the volume of extraction solution was 400 µL and the lowest concentration at 100% detection rates was 1/9 (50 µL in 450 µL) of the concentration equivalent to the 50 µL sample. Assuming that the concentration equivalent to the 50 µL sample for

the BGI test was 3.12×10^6 copies/mL, then the lowest concentration at 100% detection rates was 3.12×10^6 divided by 9, namely 3.47×10^5 copies/mL.

Therefore, we have added the sentences in the “Marked Up Manuscript - For Review Only” (Introduction, Lines 188-195): The concentration equivalent to the 50 μ L sample was the lowest concentration to achieve 100% detection rates of the 50 μ L original sample before being added to the sample preservation solution. The lowest concentration at 100% detection rates was that of the diluted sample. The pre-dilution ratios varied according to the volume of sample preservation solution (3 mL for the single sampling tests and 12 mL for the 20-in-1 pooling tests) or extraction solution (point-of-care test and rapid antigen tests) to which 50 μ L samples of known concentrations were added (Table 2, Supplementary Material File 2).

Comment 4: Table 4 is not mentioned within the text.

Answer 4: Thank you for your valuable suggestion.

Because we added Table 2 to the article, Table 4 became Table 5 accordingly. We cited Table 5 in the “Marked Up Manuscript - For Review Only” (Discussion, Lines 382): In conclusion, the study evaluated the analytical sensitivity of four SARS-CoV-2 detection strategies applied in different settings and provided helpful insights into the scientific deployment of these tests (Table 5).

Comment 5: The sentence: "Mutations can occur continuously during virus transmission..."(lane 63) should be rephrased to: "naturally occurring mutations can be selected continuously during virus transmission..."

Answer 5: Thank you for your valuable suggestion.

We have rephrased the sentence into “Naturally occurring mutations can be selected continuously during virus transmission, which is a dominating obstacle to the

prevention and control of COVID-19.” in the “Marked Up Manuscript - For Review Only” (Introduction, Lines 62-64).

September 27, 2022

Dr. Rui Zhang

Graduate School, Peking Union Medical College, Chinese Academy of Medical Sciences, Beijing, People's Republic of China;
National Center for Clinical Laboratories, Beijing Hospital
Beijing
China

Re: Spectrum02143-22R1 (Evaluation of four strategies for SARS-CoV-2 detection: characteristics and prospects)

Dear Dr. Rui Zhang:

I am delighted to inform you that your manuscript has been accepted for publication at Microbiology Spectrum.

Your manuscript has been accepted, and I am forwarding it to the ASM Journals Department for publication. You will be notified when your proofs are ready to be viewed.

Sincerely,

Sen Pei
Editor, Microbiology Spectrum

Journals Department
Supplemental Material 1-2: Accept
Supplemental Material 3: Accept